# HM3: Hierarchical Multi-Objective Model Merging for Pretrained Models

**Yu Zhou[1]   Xingyu Wu[1]\*   Jibin Wu[1,2]   Liang Feng[3]   Kay Chen Tan[1]\***

[1]Department of Data Science and Artificial Intelligence
The Hong Kong Polytechnic University, Hong Kong SAR
[2]Department of Computing, The Hong Kong Polytechnic University, Hong Kong SAR
[3]College of Computer Science, Chongqing University, Chongqing, China
`zy-yu.zhou@connect.polyu.hk   {xingy.wu, jibin.wu, kctan}@polyu.edu.hk`
`liangf@cqu.edu.cn`

## Abstract

Model merging is a technique that combines multiple large pretrained models into a single model, enhancing performance and broadening task adaptability without original data or additional training. However, most existing model merging methods focus primarily on exploring the parameter space, merging models with identical architectures. Despite its potential, merging in the architecture space remains in its early stages due to the vast search space and challenges related to layer compatibility. This paper designs a hierarchical model merging framework named HM3, formulating a bilevel multi-objective model merging problem across both parameter and architecture spaces. At the parameter level, HM3 integrates existing merging methods to quickly identify optimal parameters. Based on these, an actor-critic strategy with efficient policy discretization is employed at the architecture level to explore inference paths with Markov property in the layer-granularity search space for reconstructing these optimal models. By training reusable policy and value networks, HM3 learns Pareto optimal models to provide customized solutions for various tasks. Experimental results on language and vision tasks demonstrate that HM3 outperforms methods focusing solely on the parameter or architecture space.

## 1   Introduction

Recent advancements in large pretrained models and large language models (LLMs) have demonstrated remarkable performance and strong generalization abilities across various domains, such as natural language processing [8, 84, 68]. Open-source communities have provided many pretrained models for various data types, as well as fine-tuned versions tailored to specific tasks. However, fine-tuning large models is often a complex process that requires vast amounts of high-quality data and computational resources [28, 16]. To address the challenge of building foundational models capable of handling diverse tasks under limited computational resources, model merging has gained increasing attention [35, 77]. Model merging leverages existing pretrained models to flexibly transfer and integrate knowledge without requiring the original training data or additional model training [66, 48, 37]. This approach enables the creation of new models with stronger generalization capabilities, suited to multiple tasks and scenarios [73]. In recent years, model merging has become a simple yet powerful approach for large foundational model development, with merged models showing significant potential on the Open LLM leaderboard [44]. Current model merging methods primarily focus on merging models with the same architecture in the parameter space [58, 57]. They discard

---

*\*Corresponding authors*

39th Conference on Neural Information Processing Systems (NeurIPS 2025).

most redundant parameters, and only need to design parameter adjustment strategies in the remaining space, which often obtain moderate performance [72, 80, 27]. Thus, research in the parameter space has become quite extensive and mature [24, 39, 20, 19, 18].

However, focusing solely on merging models within the parameter space significantly limits their practical utility [1, 73]. Models with different architectures exhibit broader diversity in representation capabilities and task types [38, 43, 85], potentially expanding the performance boundaries of merged models beyond those of a single architecture. Some approaches [62, 61, 74] attempt to unify different architectures via knowledge distillation before performing parameter merging. However, these methods still operate within the parameter-space paradigm and typically incur substantial training costs in distillation, especially for LLMs. Recent work has explored architecture-level merging, such as Franken merging [22] and SOLAR 10.7B [29], which stitches different layers from LLMs. Nevertheless, merging models across different architectures presents several practical challenges [17, 57], resulting in limited research in this area. Primarily, architecture-level merging alters the computational logic of the model, necessitating the design of coordination strategies to ensure internal compatibility and seamless information flow within the new architecture. Moreover, jointly exploring both the parameter space and architecture space increases the problem's complexity [82], requiring well-defined search spaces and efficient search strategies to identify the optimal model configuration. Recently, evolutionary algorithm (EA) has been employed to search for optimal architectures [1]. However, they fail to reveal the mapping between architecture sequences and performance, making them unsuitable for handling the complex, high-dimensional problem of merging multiple models. Additionally, evolutionary processes are often one-time fusions, requiring a complete restart when faced with new problems, leading to significant computational consumption [68, 64].

To this end, merging large pretrain models in parameter and architecture spaces appears to be a promising approach, which can enhance the representational ability of the merged models while maintaining performance. However, research in this area is scarce, primarily because merging models in both spaces without careful consideration can undermine their internal compatibility, potentially causing a performance collapse. In addition, the complexity of the architecture space further increases the difficulty of model merging and reduces the efficiency of existing search methods [1]. Additionally, users may have diverse preferences and expectations for the merged model, making it crucial to weight tasks differently based on these varying preferences [34, 33, 36].

To merge models across both parameter and architecture levels and achieve efficient model merging schemes, this paper proposes a hierarchical model merging method (HM3) that builds a bridge for model merging in the parameter and architecture spaces. HM3 first defines a joint optimization problem for model merging that spans both the parameter level and the architecture level. Compared to existing methods, HM3 has also taken extra considerations on conflicts or trade-offs across tasks by extending this problem to a multi-objective optimization perspective. In HM3, we sample diverse preference vectors to decompose the multi-objective problem into multiple subproblems, and simultaneously solve them to identify approximate Pareto-optimal merged models across tasks To relieve the strong coupling between variables and the exponentially large search space of each subproblem, HM3 transforms it into a bilevel optimization problem without compromising theoretical optimality. At the architecture level, an actor-critic reinforcement learning (RL) method is designed to explore inference paths with a Markov property in the layer-granularity search space, enabling the reconstruction of these optimal models. To improve efficiency in the large discrete action space, HM3 incorporates a Wolpertinger strategy for policy discretization. Once training is achieved, the policy and value networks of this actor-critic strategy can be reused to predict optimal merging architectures and parameters for different tasks. The final approximate Pareto merged models meet different preferences based on specific needs and trade-offs. The main contributions of this paper are summarized as follows:

- We propose the hierarchical model merging method (HM3), provide the definition of the joint model merging optimization problem that spans parameter and architect space, and transform it into a bilevel optimization problem without losing theoretical optimality to relieve strong coupling and vast search space.

- The proposed HM3 is the first reusable model merging framework by integrating the current parameter-merging method and designing an actor-critic-based RL method with Wolpertinger policy discretization to guide the search, exploring the optimal model configurations in both the parameter and architecture spaces.

- We propose to incorporate a multi-objective optimization paradigm into model merging processing, which allows users to prioritize the importance of multiple tasks based on task needs by searching for the approximate Pareto front of merging strategy, enabling them to select the most suitable merged model.

## 2 Hierarchical Multi-Objective Model Merging Framework

This paper aims to jointly optimize both the parameters and architecture of pretrained models to obtain a set of approximately Pareto-optimal merged models that accommodate diverse preferences under multi-task settings. Detailed related work is provided in the appendix A. Since there is a lack of definition for architecture-level merging, we propose a unified mathematical formulation for multi-objective model merging at both the parameter and architecture levels for the first time.

**Problem Formulation and Challenge Discussion**    At the parameter level, existing works already define the merging process via optimization over $\boldsymbol{\Theta} = \{\boldsymbol{\theta}_{m_t,l_t}\}_{t=1}^T$. At the architecture level, we consider the optimization of model architecture $\alpha$ as an inference path search problem, where a search token traverses layers from multiple fine-tuned or merged models to identify an inference path with total length not exceeding $T_{\max}$. This inference path is represented by a sequence $\{(m_t, l_t)\}_{t=1}^T$, where $(m_t, l_t)$ denotes the model index and layer index selected at the $t$-th search step, and $T$ is the total path length. To this end, the unified optimization problem is defined as:

$$\max_{\boldsymbol{\Theta} \in \mathcal{P} \subseteq \mathbb{R}^{d_{\boldsymbol{\Theta}}},\, \alpha=\{(m_t,l_t)\}_{t=1}^T \in \mathcal{M}} \mathcal{F}(\boldsymbol{\Theta}, \alpha) = (f_1(\boldsymbol{\Theta}, \alpha), f_2(\boldsymbol{\Theta}, \alpha), \ldots, f_K(\boldsymbol{\Theta}, \alpha)) \tag{1a}$$

$$\text{s.t.} \quad \mathcal{C}1: \quad \boldsymbol{\Theta} = \{\boldsymbol{\theta}_{m_t,l_t} | \boldsymbol{\theta}_{m_t,l_t} = \mathcal{G}\Big(\sum_{k=1}^K \varpi_k \, \boldsymbol{\theta}_{k,l_t}\Big)\}_{t=1}^T; \tag{1b}$$

$$\mathcal{C}2: \quad \varpi_k = \begin{cases} 1, & \text{if } \boldsymbol{\theta}_{k,l_t} \text{ has the same base model as } \boldsymbol{\theta}_{m_t,l_t}; \\ 0, & \text{otherwise}; \end{cases} \tag{1c}$$

$$\mathcal{C}3: \quad |\alpha| = T \leq T_{\max}; \tag{1d}$$

$$\mathcal{C}4: \quad \sum_{t=1}^{T-1} \mathbf{1}\left[\dim_{\text{out}}(m_t, l_t; \boldsymbol{\theta}_{m_t,l_t}) \neq \dim_{\text{in}}(m_{t+1}, l_{t+1}; \boldsymbol{\theta}_{m_{t+1},l_{t+1}})\right] = 0. \tag{1e}$$

where $f_k(\cdot)$ for $k \in \{1, 2, \ldots, K\}$ denotes the performance on the $k$-th task; $\mathcal{P}$ is the parameter space; $\boldsymbol{\Theta}$ is the parameters of the merged model; $\mathcal{M} = \bigcup_{T=1}^{T_{\max}} \{\alpha = \{(m_t, l_t)\}_{t=1}^T \mid m_t \in \{1, \ldots, K\}, l_t \in \{1, \ldots, L\}\}$ is the architecture space; $\mathcal{C}1$ enforces a maximum inference path length of $T_{\max}$; and $\mathcal{C}2$ strictly ensures that the output dimension of each selected layer matches the input dimension of the next. Compared to the problem formulations of existing model merging methods, our approach extends the formulation to the architecture level. By jointly considering both space, we elevate model merging from a parameter interpolation problem to a more general structural composition problem.

In (1), $\mathcal{P}$ is constructed from multiple pretrained LLMs, which results in a high-dimensional, non-convex, and piecewise linear geometric structure. Additionally, $\mathcal{M}$ is a discrete set of cross-model, cross-layer inference paths, whose size grows exponentially with the number of models $K$ and the number of layers $L$. The **strong coupling** between $\boldsymbol{\Theta}$ and $\alpha$ leads to an extremely large and complex joint search space. Furthermore, **multi-objective function** $\mathcal{F}(\boldsymbol{\Theta}, \alpha) = (f_1, \ldots, f_K)$ exhibits non-smoothness, non-convexity, and non-differentiability under such coupled variables, making it difficult to solve for traditional convex optimization or multi-objective methods. A final challenge lies in layers from different fine-tuned models must be stitched together while preserving **dimensional consistency** across the output–input interfaces.

**Transform the Problem Formulation into A Bilevel Framework**    To address the strong coupling between $\boldsymbol{\Theta}$ and $\alpha$, we reformulate (1) as a bilevel optimization problem [51, 13]. In this problem, the upper level searches for the optimal merged parameters $\boldsymbol{\Theta}^*$ in $\mathcal{P}$, while the lower level searches for the optimal inference path $\alpha^*$ under the static environment by the converged upper-level solution $\boldsymbol{\Theta}^*$. This decomposition transforms the original joint search space of size $|\mathcal{P}| \times |\mathcal{M}|$ into two sequential subproblems with complexity $|\mathcal{P}| + |\mathcal{M}|$, thereby significantly mitigating the combinatorial explosion in the search process. The bilevel optimization problem is given as:

$$\max_{\boldsymbol{\Theta} \in \mathcal{P} \subseteq \mathbb{R}^{d_{\boldsymbol{\Theta}}}} \quad \mathcal{F}\big(\boldsymbol{\Theta}, \alpha^*(\boldsymbol{\Theta})\big) = \big(f_1(\boldsymbol{\Theta}, \alpha^*(\boldsymbol{\Theta})), f_2(\boldsymbol{\Theta}, \alpha^*(\boldsymbol{\Theta})), \ldots, f_K(\boldsymbol{\Theta}, \alpha^*(\boldsymbol{\Theta}))\big) \tag{2a}$$

$$\text{s.t.} \quad \boldsymbol{\Theta} = \{\boldsymbol{\theta}_{m_t,l_t} | \boldsymbol{\theta}_{m_t,l_t} = \mathcal{G}\Big(\sum_{k=1}^{K} \varpi_k \, \theta_{k,l_t}\Big)\}_{t=1}^{T}; \tag{2b}$$

$$\varpi_k = \begin{cases} 1, & \text{if } \boldsymbol{\theta}_{k,l_t} \text{ has the same base model as } \boldsymbol{\theta}_{m_t,l_t}; \\ 0, & \text{otherwise}; \end{cases} \tag{2c}$$

$$\alpha^*(\boldsymbol{\Theta}) \in \operatorname*{arg\,max}_{\alpha = \{(m_t, l_t)\}_{t=1}^{T} \in \mathcal{M}} \mathcal{F}\big(\boldsymbol{\Theta}, \alpha\big) \tag{2d}$$

$$\text{s.t.} \quad |\alpha| = T \leq T_{\max}, \tag{2e}$$

$$\sum_{t=1}^{T-1} \mathbf{1}[\dim_{\text{out}}(m_t, l_t; \boldsymbol{\theta}_{m_t,l_t}) \neq$$
$$\dim_{\text{in}}(m_{t+1}, l_{t+1}; \boldsymbol{\theta}_{m_{t+1}, l_{t+1}})] = 0. \tag{2f}$$

**Lemma 1** (Stackelberg Equilibrium [31, 4, Thm. 3.1])**.** *Assume the follower feasible mapping $\Omega(\boldsymbol{\Theta}) = \{\alpha \in \mathcal{M} \mid |\alpha| \leq T_{\max}, \ dim_{out} = dim_{in}\}$ is non-empty for all $\boldsymbol{\Theta}$ (since $\alpha_{base} \in \Omega$), and its graph is closed due to (A1). Under Assumption 1, the bilevel optimization problem (2) admits at least one Stackelberg equilibrium $(\boldsymbol{\Theta}^*, \alpha^*)$. The associated leader-follower payoff corresponds to a global optimum of the original problem (1). The proof of Lemma 1 is provided in the appendix B.1.*

In the appendix B.1, we further prove that the bilevel optimization problem can be modeled as a Stackelberg game, for which an equilibrium solution exists. This ensures that problem transformation does not incur any loss of optimality. In this bilevel optimization problem, the lower-level optimization searches for the optimal merged model architecture. The resulting architecture determines the length of the inference path (i.e., the number of layers to be merged). The upper-level optimization then operates on the parameter set $\{\boldsymbol{\theta}_{m_t,l_t}\}_{t=1}^{T}$ corresponding to this architecture. Consequently, the optimal architecture found by the lower level dynamically determines the dimensionality and scale of the parameter search space for the upper level. This naturally forms a hierarchical decision-making structure, i.e., first optimizing the model architecture, then optimizing the corresponding model parameters, which embodies the core hierarchical nature of the proposed HM3.

**Model the User Preference into A Multi-Objective Optimization Problem**    After mitigating the strong coupling between variables, we focus on the multi-objective property of (2). To accommodate diverse user preferences, we adopt a decomposition-based strategy that explicitly guides the solution set to cover the Pareto front boundary under controllable preference vectors. Due to the non-convexity of the search space, we employ Tchebycheff decomposition strategy, which effectively approximates non-convex Pareto fronts by transforming the original multi-objective problem into $N$ preference-weighted scalar subproblems. By solving these subproblems in parallel, we obtain a set of approximately Pareto-optimal merged models that satisfy varying user preferences.

Specifically, we begin by generating $N$ preference weight vectors $\{\boldsymbol{\lambda}^{(i)}\}_{i=1}^{N}$ from $K$-dimensional probability simplex $\Delta^K = \{\boldsymbol{\lambda} \in \mathbb{R}_+^K \mid \sum_{k=1}^{K} \lambda_k = 1\}$. In this paper, we sample from the Dirichlet distribution: $\boldsymbol{\lambda}^{(i)} \sim \text{Dirichlet}(\underbrace{1, \ldots, 1}_{K}), \quad i = 1, \ldots, N$, which ensures uniform coverage over $\Delta^K$ with an unbiased mean $\mathbb{E}[\lambda_k] = \frac{1}{K}$.

Then, we estimate the ideal point of each objective by computing the best achievable task performance across all fine-tuned LLMs: $z_k^* = \max_{(\boldsymbol{\Theta}, \alpha) \in \Omega} f_k(\boldsymbol{\Theta}, \alpha)$. Finally, for each preference vector $\boldsymbol{\lambda}^{(i)}$, the upper-level subproblem using Tchebycheff scalarization is defined as:

$$\boldsymbol{\Theta}^{(i)} = \arg\min_{\boldsymbol{\Theta} \in \mathcal{P}} \left\{ \max_{k=1,\ldots,K} \lambda_k^{(i)} \cdot \left| F_k^{\text{para}}(\boldsymbol{\Theta}) - z_k^* \right| \right\}, \tag{3}$$

where $F_k^{\text{para}}(\boldsymbol{\Theta})$ denotes the objective function value of the merged model on the $k$-th task, obtained after solving the corresponding lower-level inference path problem with fixed $\boldsymbol{\Theta}$.

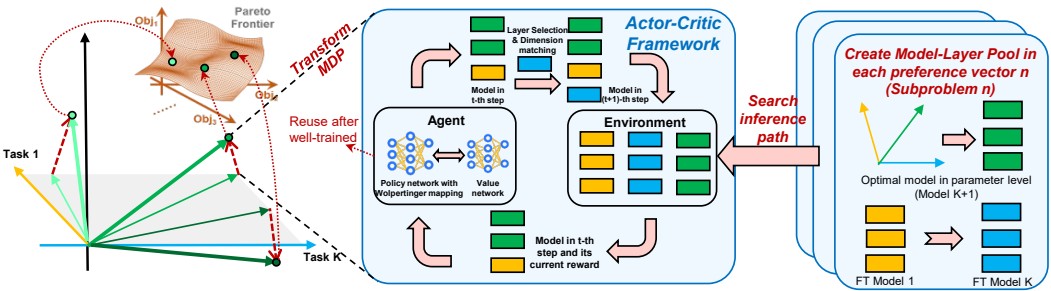

Figure 1: Illustration of architecture-level model merging. For each scalarized subproblem, we construct a model-layer candidate pool consisting of all layers from the parameter-level merged model, and $K$ fine-tuned models. Then, HM3 design an actor-critic algorithm with Wolpertinger discretization to search inference path. The final merged models approximate the Pareto front.

Once $\mathbf{\Theta}^{(i)}$ obtained, the corresponding lower-level subproblem is defined as:

$$\alpha^{(i)} = \arg \min_{\alpha \in \Omega(\mathbf{\Theta}^{(i)})} \left\{ \max_{k=1,\ldots,K} \lambda_k^{(i)} \cdot \left| f_k(\mathbf{\Theta}^{(i)}, \alpha) - z_k^* \right| \right\}, \tag{4}$$

where $\Omega(\mathbf{\Theta}^{(i)})$ denotes the feasible set of inference paths that satisfy $\mathcal{C}1$ and $\mathcal{C}2$.

Through this transformation, (1) is reduced to solving $N$ scalarized bilevel subproblems, each corresponding to a distinct preference vector $\boldsymbol{\lambda}^{(i)}$. The set of solutions to all subproblems forms an approximate Pareto-optimal set of merged models.

## 3 HM3 Algorithm

**Parameter-Level Optimization**    After generating preference vectors $\{\boldsymbol{\lambda}^{(i)} = \{\lambda_1^{(i)}, \ldots, \lambda_K^{(i)}\}\}_{i=1}^N$, we proceed to search for the optimal merged parameters in the parameter space for each preference vector. Thanks to recent advancements, parameter-level merging methods have become relatively mature and efficient. Our framework is designed to be compatible with these existing techniques, such as DARE-Ties merging method. Concretely, for $\boldsymbol{\lambda}^{(i)}$, we begin by computing the residual vector: $\delta_k = \mathbf{\Theta}_k - \mathbf{\Theta}_{\text{base}}$. We then apply the Drop-and-Rescale operation to obtain $\delta_k^{\text{DR}} = \delta_k/(1-p)$. Next, we perform the Ties Merging [72] procedure for $\boldsymbol{\lambda}^{(i)}$: removing redundant parameters from each $\delta_k^{\text{DR}}$, generating a sign-consistent aggregation mask across tasks, and merging disjoint residual fragments with consistent signs to form $\delta_k'$. Finally, the optimal parameter for the $i$-th subproblem is:

$$\mathbf{\Theta}^{(i)*} = \mathbf{\Theta}_{\text{base}} + \sum_{k=1}^K \lambda_k^{(i)} \cdot M_k^{(i)} \odot \delta_k, \tag{5}$$

where $M_k^{(i)}$ is a binary mask that controls which elements of $\delta_k$ are preserved and rescaled.

**Architecture-Level Optimization**    As discussed in Section 2, architecture-level optimization is formulated as searching for an optimal inference path across the merged model and its multiple task-specific fine-tuned models. For each $\boldsymbol{\lambda}^{(i)}$, we have already obtained the corresponding optimal merged model at the parameter level, denoted as $\mathbf{\Theta}^{(i)*}$. We assign its model index as $m = K + 1$. To further expand the search space and leverage external knowledge, we construct a model-layer candidate pool that consists of: (1) the optimal merged model $\mathbf{\Theta}^{(i)*}$ from the parameter level, and (2) all layers from the $K$ task-specific fine-tuned models used in construction of $\mathbf{\Theta}^{(i)*}$. The corresponding architecture-level search space is then updated as:

$$\mathcal{M}^{(i)} = \left\{ (m^{(i)}, l^{(i)}; \mathbf{\Theta}^{(i)}) \mid m \in \{1, \ldots, K+1\}, \, l \in \{1, \ldots, L\} \right\}. \tag{6}$$

Although prior research has demonstrated the potential of using search algorithms to optimize layer sequences and enhance model merging performance [1], the scalability of EAs suffers significantly as the number of models and layers increases [20, 18]. Moreover, EA-based approaches require

training from scratch for each preference vector and incur considerable computational cost due to population-based evaluations in every generation. These inefficiencies motivate us to revisit the nature of dynamic layer selection across multiple models [69, 67].

This process entails selecting the optimal model-layer pair at each step, considering the long-term impact of current decisions on future layer compositions and final task performance, thereby exhibiting the characteristics of a sequential decision-making problem. Furthermore, the combinatorial nature of the layer-path space, along with its discrete, structured constraints and well-defined state transitions, naturally suggests formulating the inference path search as a trajectory-aware Markov decision process (MDP). The overall process of architecture-merging is illustrated in Fig. 1. Then, we formally define its state, action, transition, and reward components and design an RL strategy to efficiently explore optimal architecture trajectories.

**1) State Space**  Since every decision in the inference path dynamically alters the feasibility of subsequent layer transitions and affects the accumulated representation distribution, we define the state to retain full trajectory history for optimal distinguishability. Formally, the state at the $t$th step is represented as a trajectory:

$$S_t = \left\{ (m_j,\ l_j,\ \boldsymbol{\theta}_{m_j,l_j}) \mid m_j \in \{1, \ldots, K+1\}, l_j \in \{1, \ldots, L\}, j = 1, \ldots, t \right\} \in \mathcal{S}, \quad (7)$$

where $m_j$ denotes the model index, $l_j$ denotes the layer index, and $\boldsymbol{\theta}_{m_j,l_j} \in \mathbb{R}^{d_\theta}$ is the corresponding parameter of the selected layer. To enable policy gradient-based learning, we use a set of learnable encoders $\psi_m$, $\psi_l$, and $\varphi$ to encode model identity, layer index, and layer parameters, respectively. The full trajectory is then embedded into a fixed-dimensional vector using a GRU encoder: $h_t = \mathrm{GRU}\left( \left[ \psi_m(m_j); \psi_l(l_j); \varphi(\boldsymbol{\theta}_{m_j,l_j}) \right]_{j=1}^t \right) \in \mathbb{R}^{d_h}$. The process on trajectory space $\mathcal{S}$ satisfies Markov property.

**2) Action Space**  At the $t$th step, the action $A_t$ is defined as selecting the next model-layer pair to transition to:

$$A_t = (m_{t+1}, l_{t+1}) \in \mathcal{A}, \quad m \in \{1, \ldots, K+1\}, l \in \{1, \ldots, L\}. \quad (8)$$

**3) Reward Function.**  The reward encourages the construction of efficient inference paths that yield high-quality multi-task performance with minimal complexity:

$$R = \sum_{k=1}^{K} \lambda_k^{(i)} f_k(\boldsymbol{\Theta},\ \boldsymbol{h}) - \beta_1 T \quad (9)$$

where the second term penalizes path length to encourage shorter and more efficient inference paths. The reward is computed only after the entire inference path is generated, and the MLP-based alignment is performed. This reward is uniformly assigned to all time steps in the inference path as $R_t = R/T, \forall t \in \{1, \ldots, T\}$. This uniform assignment is implemented to facilitate efficient storage of transitions and subsequent updates of the policy and value networks in the actor-critic framework.

**Actor-Critic Method**  To solve the MDP, HM3 employs an actor-critic-based RL strategy. In this framework, a policy network parameterized by $\mu$ outputs a probability distribution over the large discrete action space, while a value network parameterized by $\phi$ serves as a baseline to reduce the variance of gradients under sparse reward conditions. This design facilitates stable convergence of layer sequence search under limited sampling [79, 78].

The policy function $\pi_\mu(A_t \mid S_t)$ defines a stochastic policy conditioned on the current state $S_t$, representing the probability distribution over candidate actions. The distribution is modeled using a Gaussian parameterization with mean $\mu$ and variance $\xi^2$, from which actions are sampled to maximize the expected cumulative reward:

$$\max_{\mu} \ \mathbb{E}_{\pi_\mu} \left[ \sum_{t=0}^{T} \gamma^t R_t \right] = \max_{\mu} \ \mathbb{E}_{\pi_\mu} \left[ \sum_{t=0}^{T} \gamma^t \left( \sum_{k=1}^{K} \lambda_k^{(i)} f_k(\boldsymbol{\Theta}, \boldsymbol{h}_t) - \beta_1 t \right) \right], \quad (10)$$

where $\gamma \in (0, 1]$ is the discount factor, and $R_t$ denotes the reward at the $t$th step as defined earlier.

The policy network generates a continuous proto-action $\bar{A}$ following a Gaussian-distributed stochastic policy: $\bar{A} = f_{\pi(A|S;\Theta)}(S) \sim \mathcal{N}\left(\boldsymbol{\mu}_\pi(\mathbf{S}_t), \text{diag } \boldsymbol{\sigma}_\pi^2(\mathbf{S}_t)\right)$, where $f_{\pi(A|S;\Theta)}$ is the state-to-action mapping under policy $\pi$.

Since the decision variables of (2) lie in a discrete action space $\mathcal{W}$, the proto-action $\bar{A}$ must be mapped to a discrete action $\mathcal{A} \in \mathcal{W}$. Existing discretization approaches fall into two categories [55, 83]: The simple projection method, which directly selects the nearest discrete action: $\mathcal{A}^* = \arg\min_{\mathcal{A} \in \mathcal{W}} \|\mathcal{A} - \bar{A}\|$. However, this can result in suboptimal exploration and slow convergence. The greedy method, which selects the action with the highest Q-value: $\mathcal{A}^* = \arg\max_{\mathcal{A} \in \mathcal{W}} Q(\mathcal{S}, \mathcal{A})$, but this is often computationally expensive and prone to local optima.

To balance exploration and exploitation, we introduce Wolpertinger policy mapping, which improves efficiency by limiting evaluation to a local neighborhood:

$$A_t = \begin{cases} \arg\max_{\mathcal{A} \in \mathcal{W}^*(A_t)} Q_\phi(S_t, \mathcal{A}), & \text{with probability } 1 - \epsilon, \\ \mathcal{U}(\mathcal{W}^*(A_t)), & \text{with probability } \epsilon, \end{cases} \tag{11}$$

where $\mathcal{U}(\cdot)$ is a uniform distribution, and the neighborhood set $\mathcal{W}^*(A_t)$ is defined as: $\mathcal{W}^*(A_t) := \arg\min_{\substack{\mathcal{A} \in \mathcal{W} \\ |\cdot| = M'}} d(\mathcal{A}, A_t)$, with $\mathcal{W} = \{A_1, \ldots, A_\ell, \ldots, A_M\}$ denoting the full discrete action set, $d(\cdot, \cdot)$ representing Euclidean distance, and $M'$ is the number of nearest neighbors.

The value function is modeled by a state-value network $V_\phi(S_t)$, which estimates the expected cumulative reward of $S_t$: $V_\phi(S_t) = \mathbb{E}_{\pi_\mu}\left[\sum_{j=0}^{\infty} \gamma^j R_{t+j} \mid S_t\right]$, where $\sum_{j=0}^{\infty} \gamma^j R_{t+j}$ is the discounted return starting from the $t$-th step.

**Network Updates**  To stabilize policy optimization, HM3 constrains the update step size and adopts a policy gradient approach for training. The policy network is updated using the clipped surrogate objective by proximal policy optimization [49]:

$$L^{\text{CLIP}}(\mu) = \mathbb{E}_t\left[\min\left(\rho_t(\mu)\hat{A}_t, \text{ clip}(\rho_t(\mu), 1 - \epsilon, 1 + \epsilon)\hat{A}_t\right)\right], \tag{12}$$

where $\rho_t(\mu) = \frac{\pi_\mu(A_t|S_t)}{\pi_{\mu_{\text{old}}}(A_t|S_t)}$ is the importance sampling ratio between the new and old policies, and $\hat{A}_t$ is the estimated advantage. We compute $\hat{A}_t$ using the generalized advantage estimation method: $\hat{A}_t = \sum_{i=0}^{\infty}(\gamma\beta_A)^i\zeta_{t+i}$, $\zeta_t = R_t + \gamma V(S_{t+1}; \phi_{\text{iter}}) - V(S_t; \phi_{\text{iter}})$, where $\zeta_t$ is the temporal difference residual at the $t$-th step.

The value network is updated by minimizing the value loss:

$$L^{\text{VF}}(\phi) = \mathbb{E}_t\left[\left(V_\phi(S_t) - \left(V_\phi(S_t) - \hat{A}_t\right)\right)^2\right]. \tag{13}$$

The overall training objective is:

$$L(\mu, \phi) = L^{\text{CLIP}}(\mu) + c_1 L^{\text{VF}}(\phi) - c_2 H(\pi_\mu), \tag{14}$$

where $H(\pi_\mu)$ denotes the entropy of the policy, and $c_1$, $c_2$ are weighting coefficients.

**Lemma 2** (Advantage of Wolpertinger discretization). *Let $\tilde{Q}(\mathcal{S}, \mathcal{A}) = r(\mathcal{S}, \mathcal{A}) + \gamma V_\phi(\mathcal{S}')$ denote the one-step proxy score derived from the value network $V_\phi$. Consider a candidate set $\mathcal{W}^* = \{A_1, \ldots, A_\ell, \ldots, A_M\}$. Assume there exists a constant $\xi > 0$ such that: $\tilde{Q}(\mathcal{S}, A_\ell) \sim \mathcal{U}\left(\tilde{Q}(\mathcal{S}, A_s^*) - \xi, \tilde{Q}(\mathcal{S}, A_s^*) + \xi\right), \forall\ell \neq \ell'$, and that the proxy error is bounded as:*

$$\left|\tilde{Q}(\mathcal{S}, \mathcal{A}) - Q(\mathcal{S}, \mathcal{A})\right| \leq \delta, \qquad \forall \mathcal{A} \in \mathcal{W}^*. \tag{15}$$

*When $M > 1$ and $\delta < \xi\left(1 - \frac{2(2M-1)}{M \cdot 2^M}\right)$, we can confirm that Wolpertinger is expected to outperform simple nearest-neighbor projection in terms of the true Q-value. Moreover, by reducing the candidate space from $|\mathcal{W}|$ to $|\mathcal{W}^*|$, Wolpertinger achieves greater efficiency than full greedy search over all actions. The proof of Lemma 2 is provided in the appendix B.2.*

**Dimension Alignment via Statistical Matching**    To accommodate distributional shifts across layers from different models, we introduce a feed-forward MLP network that generates a scaling matrix $W_{m,l}$. The input to the MLP consists of the layer index pair $(m, l)$ and the current time step $t$, and its output is defined as:

$$W_{m,l} = \text{MLP}_\mu(m, l, t), \tag{16}$$

where $\text{MLP}_\mu$ is parameterized by $\mu$ and optimized via actor-critic method. This design is motivated by the theory of moment matching [53]. Further theoretical details are provided in the appendix B.4. It is worth that the proposed HM3 is in its early exploratory stage, and we discuss the existing limitations and possible future directions in the appendix D.

## 3.1    Experiment Setup

**Baselines**    We evaluate the proposed parameter-and-architecture hierarchical merging framework HM3[2] against three types of baselines on both language and vision tasks: fine-tuned models, three classical parameter-level merging methods, including Task Arithmetic [27], Ties-Merging [72], and DARE-Ties Merging [80], two SOTA parameter-level merging methods, including PCB Merging [19] and Consensus Merging [65], and an architecture-level merging method named EA [1].

**Benchmarks and Metrics**    For language tasks, we used LLAMA-2-7B [60], Qwen-2.5-1.5B [75], and LLAMA-2-13B [60] as backbones across four subtasks: generative task, text translation, math reasoning, and code generation. For generative tasks, we used GLUE benchmark [63] to evaluate the general capability of large pretrained models. For translation, we used WMT14, WMT16 [50], and IWSLT2017 [7] (WMT&ISWT), evaluated by the `chrf` metric as well as Xnli [15] evaluated by the `accuracy` metric. For math reasoning, we used GSM8K [12] with the `flexible match` metric, and used MathQA [3] with the `accuracy` metric. For code generation, HumanEval [9] and MBPP [5] was used with the `pass@1` and `pass@100` metric. Additionally, Qwen-2.5-1.5B was evaluated on four 3090 GPUs (24GB each), while LLaMA-2-7B and LLaMA-2-13B were evaluated on four A6000 GPUs (48GB each). All models can also be deployed on a single GPU. For vision tasks, we adopted ViT-B/32 and ViT-L/14 from CLIP [46] as backbones, and evaluated on eight datasets: DTD [11], GTSRB [52], RESISC45 [10], SUN397 [70], SVHN [45], MNIST [32], Cars [30], and EuroSAT [23], using classification accuracy. Other settings and details are summarized in the appendix C.1.

## 3.2    Performance of Multi-Task Scenario

**Merging LLAMA-2-7B LLMs**    Table 1 summarizes the performance of various merging methods across three language tasks on LLAMA-7B series LLMs. Among the fine-tuned models, WizardMath-7B [41] excelled at math due to task-specific training, while CodeLlama-7B [47] dominated code generation. Llama-2-7B-Chat [60] showed relatively balanced performance, particularly in translation. Across merging methods, Task Arithmetic provided moderate gains across tasks, whereas Ties Merging and DARE-Ties Merging achieved better trade-offs, especially in translation and code. However, EA underperformed, likely due to its unguided architecture search, which struggles to find optimal layer combinations with limited evaluations. Our proposed HM3 significantly outperformed all baselines, achieving top scores in all tasks. These results highlight the effectiveness of jointly optimizing both parameter fusion and architectural composition.

**Merging Qwen-1.5B LLMs**    To assess the robustness of HM3, we conducted merging experiments using the Qwen-2.5-1.5B series LLMs. As shown in Table 2, each fine-tuned model performed best on its own task but showed clear limitations on others, reflecting the trade-offs of single-task fine-tuning. In contrast, HM3 consistently outperformed all baselines, achieving top results in math and code, and competitive performance in translation. EA performed the worst across all tasks due to its unguided structure search. An interesting observation from Table 1 and Table 2 is that the models by HM3 sometimes outperform fine-tuned models, which are typically considered performance upper bounds for their respective tasks. We discuss this in the appendix C.2. Additionally, we conducted the experiment on LLAMA-13B, and the results and analysis are provided in the appendix C.2.

---

[2]The implementation of HM3 is available at available at this page.

Table 1: Comparison of merging methods for Llama-7B series LLMs on language tasks

| Merging Methods | General | Translation | | Math | | Code | |
|---|---|---|---|---|---|---|---|
| | Glue | WMT&ISWT | Xnli | GSM8k | MathQA | HumanEval | MBPP |
| Fine-tuned Model - Chat | 55.97 | 40.23 | 43.21 | 15.39 | 25.33 | 19.51 | 24.47 |
| Fine-tuned Model - Math | 29.32 | 34.97 | 38.93 | 45.79 | 27.09 | 20.73 | 16.96 |
| Fine-tuned Model - Code | 18.39 | 33.86 | 40.37 | 12.89 | 28.76 | 43.21 | 52.67 |
| Task Arithmetic | 35.32 | 31.30 | 32.92 | 37.83 | 17.30 | 21.36 | 22.20 |
| Ties Merging | 38.62 | 34.15 | 35.72 | 29.73 | 22.40 | 26.35 | 31.48 |
| DARE-Ties Merging | 37.03 | 33.93 | 37.47 | 38.20 | 22.70 | 28.20 | 30.11 |
| Consensus Merging | 47.52 | 37.97 | 40.74 | 37.95 | 27.99 | 30.15 | 37.93 |
| PCB Merging | 49.11 | 39.29 | 33.35 | 39.25 | 28.14 | 32.53 | 39.05 |
| EA | 21.33 | 37.51 | 27.07 | 25.51 | 22.64 | 25.17 | 16.84 |
| HM3 | **51.04** | **41.68** | **40.24** | **45.62** | **28.08** | **43.62** | **44.62** |

Table 2: Comparison of merging methods for Qwen-1.5B series LLMs on language tasks

| Merging Methods | Generative | Translation | | Math | | Code | |
|---|---|---|---|---|---|---|---|
| | Glue | WMT&ISWT | Xnli | GSM8k | MathQA | HumanEval | MBPP |
| Fine-tuned Model - Chat | 57.76 | 39.01 | 41.39 | 14.32 | 28.67 | 12.11 | 40.60 |
| Fine-tuned Model - Math | 41.95 | 23.08 | 35.36 | 32.61 | 43.33 | 13.90 | 44.05 |
| Fine-tuned Model - Code | 28.30 | 24.71 | 41.60 | 15.60 | 32.67 | 34.42 | 52.34 |
| Task Arithmetic | 42.76 | 27.40 | 30.90 | 19.73 | 37.71 | 17.63 | 21.45 |
| Ties Merging | 42.72 | 29.07 | 28.46 | 22.66 | 36.07 | 16.32 | 19.61 |
| DARE-Ties Merging | 38.25 | 27.91 | 30.87 | 20.63 | 40.33 | 19.61 | 24.84 |
| Consensus Merging | 46.42 | 29.82 | 38.09 | 23.98 | 37.84 | 22.59 | 34.08 |
| PCB Merging | 47.25 | 30.05 | 38.31 | 24.29 | 36.90 | 21.87 | 41.33 |
| EA | 29.77 | 21.36 | 23.87 | 17.40 | 35.68 | 15.33 | 23.27 |
| HM3 | **48.22** | **32.26** | **41.73** | **28.05** | **40.13** | **34.31** | **51.80** |

**Merging ViT-B/32 model**    As shown in Table 3, HM3 outperforms all baselines with an average accuracy of 66.91%. It achieves 77.21% on EuroSAT, 77.62% on SVHN, and 68.21% on GTSRB. While slightly lower on DTD, HM3 still surpasses Ties Merging and Task Arithmetic.

**Merging ViT-L/14**    Table 4 shows that HM3 consistently achieves the best results across most datasets, with 90.48% on SVHN and 83.43% on GTSRB. The overall average accuracy reaches 80.30%, significantly exceeding all other methods. The detailed analysis is in the appendix C.2.

### 3.3   Performance of Multi-Objective Model Merging

HM3 generates a diverse set of approximately Pareto-optimal merged models, enabling flexible adaptation to different user preferences. Unlike existing methods that output a single solution, HM3 provides multiple high-quality candidates. To evaluate solution quality, we compute Pareto dominance relations by pooling all solutions. A solution $x_a$ is dominated by $x_b$ if $x_b$ is no worse in all objectives and strictly better in at least one. Figure 2 shows that every baseline is dominated by at least one HM3 solution (S1–S15), demonstrating HM3's superiority in objective space. We also compare HM3 with a multi-objective evolutionary algorithm (MOEA) baseline using the hypervolume (HV)

Table 3: Performance of different model merging methods for ViT-B/32 series models on vision tasks.

| Method | Average | SUN397 | RESISC45 | SVHN | GTSRB | DTD | MNIST | Cars | EuroSAT |
|---|---|---|---|---|---|---|---|---|---|
| Task Arithmetic | 69.44 | 61.41 | 72.42 | 73.74 | 66.12 | 49.82 | 93.81 | 62.14 | 76.09 |
| Ties Merging | 69.00 | 62.34 | 71.49 | 73.68 | 62.69 | 48.52 | 96.91 | 61.06 | 75.30 |
| DARE-Ties Merging | 69.86 | 60.22 | 71.36 | 76.56 | 65.94 | 50.84 | 97.05 | 60.84 | 76.05 |
| Consensus Merging | 72.06 | 64.73 | 73.51 | 79.46 | 69.03 | 52.63 | 96.89 | 63.06 | 77.20 |
| PCB Merging | 73.80 | **63.58** | 75.71 | **82.31** | 72.57 | **54.78** | **97.42** | 64.42 | 79.63 |
| EA | 59.45 | 53.27 | 62.14 | 59.32 | 56.16 | 32.97 | 95.34 | 54.03 | 62.33 |
| HM3 | **73.83** | 63.42 | **76.27** | 82.11 | **73.11** | 54.60 | 96.85 | **64.63** | **79.66** |

Table 4: Performance of different model merging methods for ViT-L/14 series models on vision tasks.

| Method | Average | SUN397 | RESISC45 | SVHN | GTSRB | DTD | MNIST | Cars | EuroSAT |
|---|---|---|---|---|---|---|---|---|---|
| Task Arithmetic | 79.48 | 69.56 | 83.60 | 80.51 | 70.58 | 65.88 | 98.02 | 82.13 | 85.53 |
| Ties Merging | 81.28 | 68.53 | 81.89 | 87.42 | 81.72 | 58.07 | 98.89 | 84.97 | 88.77 |
| DARE-Ties Merging | 83.72 | 72.07 | 87.19 | 88.03 | 84.50 | 64.49 | **99.01** | 85.93 | 88.53 |
| Consensus Merging | 83.75 | 73.39 | 88.05 | 87.43 | 81.16 | 66.04 | 98.88 | 84.26 | 90.81 |
| PCB Merging | 85.23 | **75.04** | **88.75** | 86.46 | **86.55** | 69.13 | 98.91 | 86.01 | **91.01** |
| EA | 69.48 | 61.95 | 58.11 | 76.32 | 66.36 | 50.04 | 96.77 | 75.04 | 71.24 |
| HM3 | **85.34** | 74.76 | 88.43 | **90.02** | 85.17 | **70.21** | 98.44 | **86.33** | 89.32 |

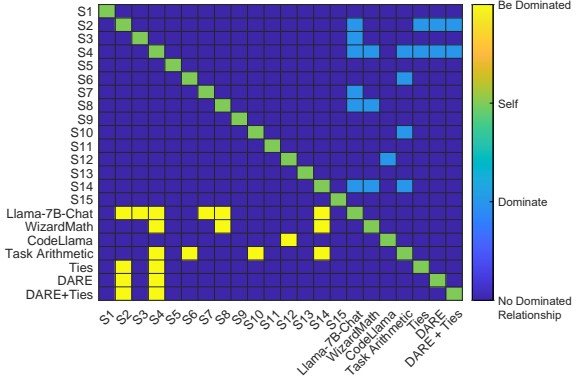

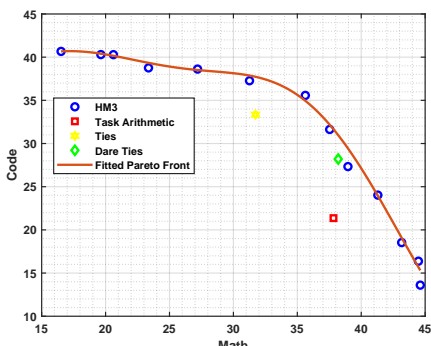

Figure 2: Illustration of metrics for different merging methods, where S1 represents solution1 obtained by HM3.

Figure 3: The illustration of different model merging methods in the math reasoning and code generation tasks.

metric [25, 56], which reflects both convergence and diversity. HM3 achieves an HV of 1.8120, significantly higher than MOEA's 1.5111, highlighting the limitations of unguided evolutionary search in complex multi-objective scenarios. Detailed analysis is provided in the appendix C.3. The effectiveness of HM3 on different numbers of objectives is provided in the appendix C.4.

### 3.3.1 Ablation Study

To evaluate the effectiveness of jointly optimizing parameter and architecture spaces, we conduct ablation studies on three variants: (i) HM3, (ii) HM3 w.o. arch (no architecture optimization), and (iii) HM3 w.o. para (no parameter optimization), with results shown in Table 6 in the appendix C.5. In the single-objective setting, HM3 outperforms both ablated versions on all tasks, especially in code generation, highlighting the synergy between parameter and architecture optimization. In the multi-objective setting, HM3 achieves the highest HV score, followed by HM3 w.o. arch, while HM3 w.o. para performs the worst. This demonstrates that parameter optimization is critical for overall performance, and architecture optimization further enhances solution quality. Detailed analysis is provided in the appendix C.5. We also analyze the **computational cost** of HM3 compared to the conventional pretraining and fine-tuning paradigm in the appendix C.6. Additionally, **convergence analysis of RL** is provided in the appendix C.7.

## 4 Conclusion

In this paper, we propose HM3, a hierarchical model merging framework that jointly optimizes parameter and architecture spaces. By leveraging an actor-critic strategy and preference-guided multi-objective optimization, HM3 efficiently generates customized, high-performing merged models. Extensive experiments on translation, math reasoning, and code generation tasks demonstrate HM3's superiority over existing methods. The framework learns Pareto-optimal solutions tailored to diverse user preferences, offering a flexible and scalable approach to model merging. Future work will explore applying HM3 to larger-scale pretrained models for broader generalization and adaptability.

## Acknowledgment

This work was partially supported by National Natural Science Foundation of China under Grant U21A20512 and in part by the Research Grants Council of the Hong Kong SAR under Grant No. C5052-23G, Grant PolyU 15229824, Grant PolyU 15218622, and Grant PolyU 15215623. This work was also partially supported the Research Grants Council of the Hong Kong SAR (Grant No. PolyU15217424, PolyU25216423), and The Hong Kong Polytechnic University (Project IDs: P0043563). This work was also in part by the Natural Science Foundation of Chongqing (Innovation and Development Joint Fund) under Grant CSTB2025NSCO-LZX0014.

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

# Appendix of HM3

# A  Comprehensive Related Work about HM3

## A.1  Model Merging

Model merge refers to combining the parameters and features of multiple large pretrained models to generate a unified model that can perform better across multiple tasks. Existing model merging approaches rely on task vectors constructed from fine-tuned models and their common base model. These approaches typically perform parameter-level interpolation (e.g., Task Arithmetic [27], TIES merging [72], PCB Merging [19], CAT Merging [54], and Consensus Merging [65]) or apply parameter manipulation strategies such as drop and rescale in DARE [80]. Formally, given a base model with parameters $\boldsymbol{\Theta}_{\text{base}}$, and $K$ task-specific models fine-tuned from it with parameters $\{\boldsymbol{\Theta}_1, \boldsymbol{\Theta}_2, \ldots, \boldsymbol{\Theta}_K\}$, the model merging process can be expressed as [14]:

$$\boldsymbol{\Theta}_{\text{merge}} = \mathcal{G}(\boldsymbol{\Theta}_{\text{base}}, \boldsymbol{\Theta}_1, \boldsymbol{\Theta}_2, \ldots, \boldsymbol{\Theta}_K) \tag{17}$$

where $\boldsymbol{\Theta}_{\text{merge}}$ denotes the parameters of the merged model, and $\mathcal{G}(\cdot)$ represents the merging method.

These methods have demonstrated significant improvements in the performance of merged models. In particular, recent works have introduced evolutionary search algorithms, such as evolutionary algorithms (EAs), to enhance model merging. For example, GENOME [81] employs one of the classical EAS, named differential evolution, to evolve new models within a shared architectural weight space through crossover, mutation, and selection operations, and further performs ensemble inference. Similarly, Evolver [20] directly applies EA to the weight spaces of multiple fine-tuned models, mutating and crossing their parameter vectors to select higher-performing combinations, thereby achieving parameter fusion without gradient-based fine-tuning. Both methods emphasize low-cost and high-efficiency strategies that yield competitive performance across different scenarios. However, they primarily focus on adjusting parameter configurations within a fixed architecture. In practice, models with diverse architectures may exhibit stronger representation capacities and potentially extend performance beyond the limits of a single-structure model under multi-task scenarios. Motivated by this, Akiba et al. [1] recently explored the use of EA to search for optimal architectures in model merging. While promising, this approach faces scalability issues: as model size increases, the architecture search space becomes substantially more complex, often resulting in performance degradation. Furthermore, EA is population-based and requires expensive evaluations in each iteration. They are also typically designed for one-shot merging, which means that the search must be restarted from scratch for every new task. This leads to prohibitively high computational costs. In this work, we unify the strengths of both parameter-level and architecture-level merging by designing an efficient joint framework. Importantly, we train a reusable model that can generalize to new tasks without requiring full re-search from scratch.

**Multi-Objective in Model Merging**  Existing methods typically rely on the model designer's domain knowledge or intuitive understanding to manually determine these weights, resulting in a single trade-off solution for the merged model. However, task preferences may differ across users, or even for the same user at different times, thereby demanding merged models that reflect diverse preferences. Recent works [34, 33] have begun to explore the flexibility of multi-objective optimization in assigning weights across tasks. Nonetheless, these efforts remain in their early stages and often fail to fully exploit the nature of multi-objective trade-offs, falling short of achieving high-quality Pareto-optimal merged models. In this paper, we design a multi-objective strategy to obtain approximate Pareto-optimal parameters and architectures that meet different preferences.

## A.2  Multi-objective Optimization

### A.2.1  Definition

Generally, a multi-objective optimization problem can be formulated as:

$$\min f(\boldsymbol{x}) = (f_1(\boldsymbol{x}), f_2(\boldsymbol{x}), \ldots, f_K(\boldsymbol{x})) \quad s.t. \quad \boldsymbol{x} \in X, \tag{18}$$

where $\boldsymbol{x} = (x_1, x_2, \ldots, x_d)$ is a decision vector, and $f(\cdot) : X \rightarrow Y$ represents $k$ objective functions. Here, $X$ denotes the decision space, and $Y$ denotes the objective space. To compare the quality of solutions obtained by the multi-objective problem, the concept of Pareto dominance is introduced.

**Pareto dominance** Given two solutions $\boldsymbol{x}_1$ and $\boldsymbol{x}_2$ belonging to $X$, $\boldsymbol{x}_1$ is said to Pareto dominate $\boldsymbol{x}_2$ (denoted as $\boldsymbol{x}_1 \prec \boldsymbol{x}_2$) if and only if the following two conditions are satisfied:

1. For all objectives $i \in \{1, 2, \ldots, K\}$, $f_i(\boldsymbol{x}_1) \leq f_i(\boldsymbol{x}_2)$, meaning that $\boldsymbol{x}_1$ is not worse than $\boldsymbol{x}_2$ in every objective.

2. There exists at least one objective $j \in \{1, 2, \ldots, m\}$ such that $f_j(\boldsymbol{x}_1) < f_j(\boldsymbol{x}_2)$, indicating that $\boldsymbol{x}_1$ is strictly better than $\boldsymbol{x}_2$ in at least one objective.

A solution $\boldsymbol{x}^* \in X$ is considered Pareto optimal if no other solution $\boldsymbol{x} \in X$ Pareto dominates $\boldsymbol{x}^*$. The set of all Pareto optimal solutions is known as the Pareto set:

$$PS = \{\boldsymbol{x} \in X \mid \nexists\, \boldsymbol{x}' \in X, \boldsymbol{x}' \prec \boldsymbol{x}\} \tag{19}$$

The collection of objective vectors corresponding to the Pareto set is referred to as the Pareto front. Multi-objective optimization aims to approximate the Pareto set by identifying solutions that achieve both strong convergence and a diverse spread within the objective space.

In multi-objective optimization methods, since the true Pareto optimal solution set is unknown, we employ the commonly used metric called hypervolume (HV) [56] to comprehensively assess the diversity and convergence of the generated approximate Pareto optimal solution set. Let a point set $P \subset \mathbb{R}^d$ and a reference point $\mathbf{r} \in \mathbb{R}^d$, where $d = 3$ is the number of optimization objectives. The HV of the set $P$ is computed as follows:

$$\mathrm{HV}(P, \mathbf{r}) = \mathcal{L}_e \left( \bigcup_{\mathbf{p} \in P} \{\mathbf{q} \mid \mathbf{p} \preceq \mathbf{q} \preceq \mathbf{r}\} \right) \tag{20}$$

where $\mathcal{L}_e(\cdot)$ represents the Lebesgue measure of a set: $\mathcal{L}_e(\mathcal{S}) = \int_{\mathbf{s} \in \mathcal{S}} \mathbf{1}_{\mathcal{S}}(\mathbf{s})\, d\mathbf{s}$ Here, $\mathbf{1}_{\mathcal{S}}$ is the characteristic function of the objective space $\mathcal{S}$. If $\mathbf{s} \in \mathcal{S}$, then $\mathbf{1}_{\mathcal{S}}(\mathbf{s}) = 1$; otherwise, $\mathbf{1}_{\mathcal{S}}(\mathbf{s}) = 0$. In the calculation of HV, the non-dominated solutions obtained by each algorithm are normalized using the same reference set, and the reference point is typically set at $(1.1, 1.1)$. It is important to note that a larger HV indicates a better approximation of the Pareto optimal solution set and, consequently, improved performance of the corresponding multi-objective optimization method.

As for multi-objective optimization in model merging, there are two early explorations. The first paper [34] introduced a novel method called model merging with amortized Pareto fronts, which approximated evaluation metrics using a quadratic surrogate model derived from a set of pre-selected scaling coefficients. However, while this approach primarily focuses on reducing computational complexity, it does not thoroughly explore how to accurately obtain the Pareto-optimal merged model. The second paper [33] employed parallel multi-objective Bayesian optimization to systematically explore the parameter space for optimal merging configurations. However, these works are only in the early stages of exploration. They merely use multi-objective optimization methods to facilitate model merging, but do not fully consider the multi-objective and multi-task characteristics inherent in the models during the merging process.

# B   Detail of the proposed HM3

## B.1   The Proof of Problem Transformation

To handle the $K$-dimensional vector-valued objective $\boldsymbol{f} = (f_1, \ldots, f_K)$, we adopt a standard linear scalarization approach. Specifically, for a given task preference vector $\boldsymbol{\lambda} \in \Delta^K$ sampled from a Dirichlet distribution, the scalarized objective is defined as:

$$F_{\boldsymbol{\lambda}}(\boldsymbol{\Theta}, \alpha) := \sum_{k=1}^{K} \lambda_k f_k(\boldsymbol{\Theta}, \alpha). \tag{21}$$

We write $F := F_{\boldsymbol{\lambda}}$ for brevity. Solving the Stackelberg game for each $\boldsymbol{\lambda}$ produces a set of solutions that approximates the Pareto front.

**Assumption 1** (Compactness and Continuity). *1. The parameter space $\mathcal{P} \subset \mathbb{R}^{d_\theta}$ is nonempty and compact. The architecture space $\mathcal{M}$ is finite and contains at least one feasible base path $\alpha_{\mathrm{base}}$.*

2. *For any* $\mathbf{\Theta} \in \mathcal{P}$, *the follower's feasible set*

$$\Omega(\mathbf{\Theta}) := \left\{ \alpha \in \mathcal{M} \,\middle|\, |\alpha| \leq T_{\max}, \; dim_{out}(m_t, l_t; \boldsymbol{\theta}_{m_t, l_t}) = dim_{in}(m_{t+1}, l_{t+1}; \boldsymbol{\theta}_{m_{t+1}, l_{t+1}}), \; \forall t \right\} \tag{22}$$

*is nonempty (since* $\alpha_{\text{base}} \in \Omega(\mathbf{\Theta})$*) and has a closed graph.*

3. *The scalarized utility* $F(\mathbf{\Theta}, \alpha)$ *is jointly continuous in* $(\mathbf{\Theta}, \alpha)$.

*Proof.* **(a) Follower-level solution existence.** Since $\mathcal{M}$ is finite, the constrained set $\Omega(\mathbf{\Theta}) \subseteq \mathcal{M}$ is finite and nonempty for any fixed $\mathbf{\Theta} \in \mathcal{P}$. Hence, the follower-level optimization

$$\alpha^*(\mathbf{\Theta}) := \arg \max_{\alpha \in \Omega(\mathbf{\Theta})} F(\mathbf{\Theta}, \alpha) \tag{23}$$

admits at least one solution. Furthermore, by Berge Maximum Theorem [6], the best-response mapping $\alpha^*(\mathbf{\Theta})$ is upper hemicontinuous with compact (finite) values due to the closed graph property and continuity of $F$.

**(ii) Continuity of the leader's objective.** Define the upper-level objective:

$$\bar{F}(\mathbf{\Theta}) := \max_{\alpha \in \Omega(\mathbf{\Theta})} F(\mathbf{\Theta}, \alpha). \tag{24}$$

Because $\alpha^*(\mathbf{\Theta})$ is upper hemicontinuous and $F$ is continuous, it follows from [2] that $\bar{F}$ is continuous on the compact domain $\mathcal{P}$. Therefore, by Weierstrass' Theorem, there exists a maximizer $\mathbf{\Theta}^* \in \arg \max_{\mathbf{\Theta} \in \mathcal{P}} \bar{F}(\mathbf{\Theta})$.

**(iii) Equilibrium construction.** Select any $\alpha^* \in \alpha^*(\mathbf{\Theta}^*)$. Then the pair $(\mathbf{\Theta}^*, \alpha^*)$ satisfies the definition of a Stackelberg equilibrium [31, Def. 2.2], and achieves the same optimal value as the original joint objective. □

**Remark.** Assumption (A2) is typically ensured in practice by guaranteeing at least one dimension-compatible base path (e.g., through 1×1 projections when needed). Each choice of $\boldsymbol{\lambda}$ induces a scalarized subproblem, and the collection of corresponding equilibria approximates the Pareto front.

## B.2 The Proof of Wolpertinger Policy in Actor-Critic Framework

Let the discrete candidate action set generated by the Wolpertinger policy be denoted as $\mathcal{W}^* = \{A_1, A_2, \ldots, A_M\}$, which contains the $M$ nearest neighbors in Euclidean distance of the continuous proto-action $\hat{A} \in \mathbb{R}^d$. Among them, define $A_{\ell^*}$ to be the nearest discrete action to $\hat{A}$, i.e., the one selected by the simple projection method:

$$A_{\ell^*} = \arg \min_{A \in \mathcal{W}^*} \|A - \hat{A}\|_2. \tag{25}$$

Assume the action-value function $Q(\mathcal{S}, A)$ under fixed state $\mathcal{S}$ satisfies the following statistical assumptions:

- For all $\ell \neq \ell^*$, the values $Q(\mathcal{S}, A_\ell)$ are i.i.d. samples from a uniform distribution:

$$Q(\mathcal{S}, A_\ell) \sim \mathcal{U}(Q(\mathcal{S}, A_{\ell^*}) - \xi, \; Q(\mathcal{S}, A_{\ell^*}) + \xi), \tag{26}$$

where $\xi > 0$ is a fixed constant.

- The value of the nearest action $A_{\ell^*}$ is set as the reference:

$$Q(\mathcal{S}, A_{\ell^*}) = q_0. \tag{27}$$

Let $A_w^*$ be the action selected by the Wolpertinger strategy, i.e., the one in $\mathcal{W}^*$ with the maximum Q-value:

$$A_w^* = \arg \max_{A \in \mathcal{W}^*} Q(\mathcal{S}, A). \tag{28}$$

Then, the expected Q-value of the selected action is:

$$\mathbb{E}[Q(\mathcal{S}, A_w^*)] = q_0 + \xi \left( 1 - \frac{2(2M - 1)}{M \cdot 2^M} \right). \tag{29}$$

Let $X_1, X_2, \ldots, X_{M-1}$ denote the i.i.d. uniform random variables representing the Q-values of the other $M - 1$ candidates:

$$X_i \sim \mathcal{U}(q_0 - \xi, \ q_0 + \xi), \quad i = 1, \ldots, M - 1. \tag{30}$$

Let $X_{\max} = \max\{X_1, \ldots, X_{M-1}\}$. Then the probability density function (PDF) of $X_{\max}$ is:

$$f_{X_{\max}}(x) = (M - 1) \cdot \frac{1}{2\xi} \left( \frac{x - (q_0 - \xi)}{2\xi} \right)^{M-2}, \quad x \in [q_0 - \xi, \ q_0 + \xi]. \tag{31}$$

The expected maximum is:

$$\mathbb{E}[X_{\max}] = \int_{q_0-\xi}^{q_0+\xi} x \cdot f_{X_{\max}}(x) \, dx = q_0 + \xi \cdot \left( 1 - \frac{2(2M - 1)}{M \cdot 2^M} \right). \tag{32}$$

Note that if $X_{\max} > q_0$, then the maximum selected action $A_w^*$ will not be $A_{\ell^*}$, but one of the other candidates. If all $X_i < q_0$, then $A_{\ell^*}$ is still chosen. Therefore, the expected Q-value of the Wolpertinger-selected action is:

$$\mathbb{E}[Q(\mathcal{S}, A_w^*)] = \mathbb{E}[\max\{q_0, \ X_1, \ldots, X_{M-1}\}] = \mathbb{E}[\max\{q_0, \ X_{\max}\}]. \tag{33}$$

By integrating over the support and using order statistics of uniform distributions, the final result is:

$$\mathbb{E}[Q(\mathcal{S}, A_w^*)] = q_0 + \xi \left( 1 - \frac{2(2M - 1)}{M \cdot 2^M} \right). \tag{34}$$

The proxy estimation error is bounded by:

$$\left| \tilde{Q}(\mathcal{S}, A) - Q(\mathcal{S}, A) \right| \leq \delta, \qquad \forall A \in \mathcal{W}^*. \tag{35}$$

Then the expected Q-value of the Wolpertinger-selected action satisfies:

$$\mathbb{E}\left[Q(\mathcal{S}, A_w^*)\right] \geq Q(\mathcal{S}, A_s^*) + \xi \left( 1 - \frac{2(2M - 1)}{M \cdot 2^M} \right) - \delta. \tag{36}$$

When $M > 1$ and $\delta < \xi \left( 1 - \frac{2(2M-1)}{M \cdot 2^M} \right)$, we obtain:

$$\mathbb{E}\left[Q(\mathcal{S}, A_w^*)\right] > Q(\mathcal{S}, A_s^*), \tag{37}$$

which confirms the superiority of the Wolpertinger policy even in the presence of bounded proxy approximation error.

### B.3 Algorithm Description

Based on the MDP, the execution of the RL is the following stages: **Stage 1: Input and initialization:** The algorithm begins by sampling preference vectors, each corresponding to a decomposed subproblem in the multi-objective framework. For each preference vector, an existing parameter-level merging method is applied to obtain an initial merged model parameter. Meanwhile, the parameters of the policy, value, and the MLP network for alignment are initialized. **Stage 2: Trajectory collection and MLP alignment:** For each preference vector, an inner loop is executed in RL. In each iteration, the parameter-level merged model and the fine-tuned models are evaluated to compute the reward and stored in the trajectory buffer. Then, at each step, the current trajectory-encoded state is used to generate a proto-action, which is discretized via the Wolpertinger policy to select the next action. The new state is converted, and the transition is recorded in the buffer. Once a complete path is collected, the algorithm performs MLP alignment. **Stage 3: Reward computation and network update:** After the alignment, the reward is computed and then is assigned to all time steps in the trajectory. Once the iteration number exceeds, the algorithm enters the network update phase. In this phase, a mini-batch is sampled from the buffer to compute GAE and target returns. The policy, value network, and MLP alignment network are updated. The entire process continues until $Max_{iter}$ is reached, yielding the

---
**Algorithm 1** HM3 in the architecture space
---

1: **Input:** A set of preference vectors $\{\boldsymbol{\lambda}^1, \boldsymbol{\lambda}^2, \ldots, \boldsymbol{\lambda}^N\}$ and their corresponding optimal merged models at the parameter space.
2: **for** each preference vector $\boldsymbol{\lambda}^n$ in $\{\boldsymbol{\lambda}^1, \boldsymbol{\lambda}^2, \ldots, \boldsymbol{\lambda}^N\}$ **do**
3:     Input $K$ fine-tuned models and the optimal merged model in the parameter space corresponding to $\lambda_i$.
4:     Initialize the parameters of policy network as $\mu_0$, of value network as $\phi_0$, of MLP network as $\mathrm{MLP}_{\mu 0}$.
5:     **for** each iteration $iter = 1, 2, \ldots, Max\_iter$ **do**
6:         Sample the current policy $\pi_{\mu_{iter}}(A_t|S_t)$ by interacting with the environment to generate a trajectory of length $T$ as $\{S_t, A_t, R_t, S_{t+1}\}_{t=1}^T$.
7:         Obtain the state $S_t = (m_t, l_t)$.
8:         Select the action $A_t = (m_{t+1}, l_{t+1})$.
9:         Calculate the reward $R_t$ based on $A_t$ and the merged model in the $t$-th step.
10:        Compute the advantage function $\hat{A}_t$ and $\hat{G}_t$.
11:        Update the policy network by maximizing $L^{CLIP}(\mu) = \mathbb{E}_t\left[\min\left(\rho_t(\mu)\hat{A}_t, \mathrm{clip}(\rho_t(\mu), 1-\epsilon, 1+\epsilon)\hat{A}_t\right)\right]$.
12:        Update the value network by minimizing $L^{VF}(\phi) = \mathbb{E}_t\left[\left(V_\phi(S_t) - \hat{G}_t\right)^2\right]$.
13:        Compute the scaling matrix $W_{m,l}$, and update the MLP network parameters $\mathrm{MLP}_\mu$.
14:     **end for**
15: **end for**
16: **Output:** Optimal policy network parameterized $\mu^*$ and the optimal inference path (i.e., the optimal sequence of actions) corresponding to the value network.

---

final policy network and the optimal inference paths. The well-trained networks can be reused for new tasks or incorporated into model pools.

The overall algorithm of HM3 is summarized as Algorithm 1. It begins by taking in a collection of preference vectors and the associated best-merged models from the parameter space (Line 1). Each preference vector introduces the fine-tuned models and the corresponding best-merged model (Line 3). The initial parameters for the policy, value network, and MLP network are set up (Line 4). During each loop, the policy network interacts with the environment, creating a trajectory composed of state, action, and reward information (Line 6). The state at the current step and the chosen action are determined next (Lines 7-8). Subsequently, the reward for the current action is calculated based on the state within the merged model (Line 9). These rewards are then utilized to compute the advantage and target values (Line 10). The algorithm adjusts the policy network by enhancing the policy loss and refines the value network by minimizing the value loss (Lines 11-12). The process also involves calculating the scaling matrix and fine-tuning the MLP network through PPO (Line 13). Finally, it outputs the final set of optimized policy parameters and the sequence of actions that represent the optimal inference path tied to the value network (Line 16).

After obtaining the Pareto-optimal models, HM3 assumes users typically do not provide explicit preferences and supports two practical modes: (i) Offline preference sampling: We uniformly sample preference vectors to approximate the Pareto front. Users can later select a model matching their needs (no input required). (ii) Optional user preference injection: If a user specifies a preference (e.g., prioritizing translation), we select the closest model on the front or conduct a targeted search, enabling both automated and interactive use.

### B.4 Detail Analysis of Dimension Alignment via Statistical Matching

To accommodate distributional shifts across layers from different models, we introduce a feed-forward MLP network that generates a scaling matrix $W_{m,l}$. The input to the MLP consists of the layer index pair $(m, l)$ and the current time step, and its output is defined as:

$$W_{m,l} = \mathrm{MLP}_\mu(m, l, t), \tag{38}$$

where $\mathrm{MLP}_\mu$ is parameterized by $\mu$ and optimized via actor-critic method.

This design is motivated by the theory of statistical moment alignment. Suppose the hidden representations from the source and target layers follow Gaussian distributions:

$$z_{\text{src}} \sim \mathcal{N}(\mu_{\text{src}}, \Sigma_{\text{src}}), \quad z_{\text{tgt}} \sim \mathcal{N}(\mu_{\text{tgt}}, \Sigma_{\text{tgt}}). \tag{39}$$

The squared 2-Wasserstein distance between them is:

$$W_2^2 = \|\mu_{\text{src}} - \mu_{\text{tgt}}\|_2^2 + \text{Tr}\left(\Sigma_{\text{src}} + \Sigma_{\text{tgt}} - 2(\Sigma_{\text{tgt}}^{1/2} \Sigma_{\text{src}} \Sigma_{\text{tgt}}^{1/2})^{1/2}\right), \tag{40}$$

and the optimal affine transformation $T(z) = A_{\text{opt}} z + b$ is given by:

$$A_{\text{opt}} = \Sigma_{\text{tgt}}^{1/2} \Sigma_{\text{src}}^{-1/2}, \quad b = \mu_{\text{tgt}} - A_{\text{opt}} \mu_{\text{src}}. \tag{41}$$

This whitening followed by coloring transformation achieves exact alignment of first- and second-order statistics. To enable end-to-end learning, we approximate this process using the deep CORAL loss [53]:

$$\mathcal{L}_{\text{Deep-CORAL}} = \|C(H'_{\text{src}}) - C(H_{\text{tgt}})\|_F^2, \tag{42}$$

where $H'_{\text{src}} = W_{m,l}\left((H_{\text{src}} - \mu_{\text{src}}) \oslash \sigma_{\text{src}}\right) + b_{m,l}$, and $C(\cdot)$ denotes the empirical covariance matrix. If mean alignment is also desired, we add a mean alignment loss:

$$\mathcal{L}_\mu = \|\mu_{\text{src}} - \mu_{\text{tgt}}\|_2^2. \tag{43}$$

The final training objective for the MLP-based alignment layer becomes:

$$z'_{\text{src}} = W_{m,l} \cdot \frac{z_{\text{src}} - \mu_{\text{src}}}{\sqrt{\text{diag}(\Sigma_{\text{src}})}} + b_{m,l}, \qquad \min_{W_{m,l}, b_{m,l}} \mathcal{L}_{\text{Deep-CORAL}} + \lambda \mathcal{L}_\mu. \tag{44}$$

This allows the MLP to approximate the theoretically optimal mapping $(A_{\text{opt}}, b)$ via gradient descent, thereby providing robust statistical alignment for seamless composition of heterogeneous transformer layers.

## C  Additional Results

### C.1  Detail of Experimental Setup

The core objective of this study is to design and implement an efficient and multitask-adaptive model merging framework. To verify the generality and performance of the proposed merging method, we apply it to three popular series LLMs, namely the fine-tuned models based on Qwen-2.5-1.5B [3] [59], Llama-2-7B [4] [60] and Llama-2-13B [5]. Specifically, for Llama-2-7B [60], we include three fine-tuned models: Llama-7B-Chat [6] for text translation [60], WizardMath-7B [7] for mathematical reasoning [40], and CodeLlama-7B [8] for code generation [47]. Similarly, for Qwen-2.5-1.5B [59], we include three fine-tuned models: Qwen-2.5-1.5B-Instruct [9] for text translation [59], Qwen-2.5-Code-1.5B [10] for code generation [26], and Qwen-2.5-Math-1.5B [11] for mathematical reasoning [76]. As for Llama-2-13B, we include three fine-tuned models: WizardLM-13B [12] for text translation [71], WizardMath-13B [13] for mathematical reasoning [40], and WizardCoder-Python-13B [14] for code generation [42].

- Llama-7B-Chat: https://huggingface.co/meta-llama/Llama-2-7b-chat-hf;

---

[3] https://huggingface.co/Qwen/Qwen2.5-1.5B

[4] https://huggingface.co/meta-llama/Llama-2-7b-hf

[5] https://huggingface.co/meta-llama/Llama-2-13b-hf

[6] https://huggingface.co/meta-llama/Llama-2-7b-chat-hf

[7] https://huggingface.co/WizardLMTeam/WizardMath-7B

[8] https://huggingface.co/codellama/CodeLlama-7b-hf

[9] https://huggingface.co/Qwen/Qwen2.5-1.5B-Instruct

[10] https://huggingface.co/Qwen/Qwen2.5-Coder-1.5B

[11] https://huggingface.co/Qwen/Qwen2.5-Math-1.5B

[12] https://huggingface.co/WizardLMTeam/WizardLM-13B-V1.2

[13] https://huggingface.co/vanillaOVO/WizardMath-13B-V1.0

[14] https://huggingface.co/WizardLMTeam/WizardCoder-Python-13B-V1.0

- WizardMath-7B: https://huggingface.co/WizardLMTeam/WizardMath-7B;

- CodeLlama-7B: https://huggingface.co/codellama/CodeLlama-7b-hf;

- LLama-2-7B: https://huggingface.co/meta-llama/Llama-2-7b-hf.

By leveraging these fine-tuned LLMs, we can utilize our proposed HM3 to merge LLMs across multiple tasks. To evaluate the performance of the merged LLMs by HM3, we perform three tasks, including language translation, mathematical reasoning, and code generation. To achieve these evaluations quickly and efficiently, we employed two popular large model evaluation packages: lm-evaluation-harness [21] for text translation and mathematical reasoning tasks and big code-evaluation-harness for code generation tasks. These evaluation packages can be found at the following link:

- lm-evaluation-harness: https://github.com/EleutherAI/lm-evaluation-harness;

- bigcode-evaluation-harness: https://github.com/bigcode-project/bigcode-evaluation-harness.

To further demonstrate the effectiveness and superiority of our method compared to other model merging methods, we utilized the mergekit package [22] to merge models by using several merging methods, including Task Arithmetic, Ties, DARE-Ties and EA [1]. The mergekit package can be found at the following link:

- mergekit: https://github.com/arcee-ai/mergekit

Additionally, for HM3, $Max_{iter}$ is 1000 and the discount factor $\gamma$ is configured to 0.990. We split the dataset where 70% is used for RL inference evaluation, while the 30% is reserved for the evaluation of the obtained merged model. Then, we introduce the specific datasets for generative, text translation, math reasoning, and code generation tasks, as well as their metrics.

### C.1.1 Generative Tasks

We use GLUE benchmark [63], a widely used benchmark for natural language understanding comprising nine sentence/sentence-pair classification tasks (i.e., CoLA, SST-2, MRPC, STS-B, QQP, MNLI, QNLI, RTE, WNLI). These tasks, drawn from established datasets, cover a range of sizes, genres, and difficulties, offering a broad and challenging evaluation of language understanding.

### C.1.2 Text Translation Tasks

To evaluate the multilingual translation capabilities of LLMs, we leveraged a set of translation tasks in the lm-evaluation-harness package, including WMT14[15], WMT16[16] [50], and IWSLT2017 [7], and Xnli [15]. These tasks evaluate the model's translation accuracy and fluency across diverse language pairs. For the first translation tasks, we use the "chrf" metric, which measures translation quality based on character n-gram precision and recall. For the final task, we use accuracy as the evaluation metric instead.

### C.1.3 Math Reasoning Task

In this paper, we use MathQA [3] and GSM8K [12] for the evaluation of the math reasoning capability of the obtained models. MathQA [3] is a large-scale benchmark of roughly 37K English multiple-choice math word problems covering a broad range of mathematical topics. It also provides operation programs aligned with problems from the AQuA dataset. Model performance on MathQA is reported as accuracy. GSM8K [12] is a dataset meticulously designed for mathematical problem-solving tasks, comprising over 8,000 high-quality problems that span from basic arithmetic to complex algebra. The primary objective of this dataset is to evaluate the model's reasoning and computational abilities when tackling structured mathematical problems. For evaluating the GSM8K dataset, we employ the "flexible match" metric, which allows for minor variations in the final answer.

---

[15]https://www.statmt.org/wmt14/translation-task.html
[16]https://www.statmt.org/wmt16/translation-task.html

Table 5: Comparison of merging methods for Llama-2-13B series LLMs on language tasks

| Merging Methods | General | Translation | | Math | | Code | |
|---|---|---|---|---|---|---|---|
| | Glue | WMT&ISWT | Xnli | GSM8k | MathQA | HumanEval | MBPP |
| Fine-tuned Model - Chat | 67.05 | 43.29 | 47.11 | 33.82 | 22.67 | 21.32 | 22.56 |
| Fine-tuned Model - Math | 45.08 | 27.02 | 41.23 | 55.74 | 31.33 | 19.21 | 20.83 |
| Fine-tuned Model - Code | 19.27 | 32.49 | 42.61 | 14.75 | 27.09 | 51.82 | 29.63 |
| Task Arithmetic | 35.24 | 29.16 | 37.33 | 27.74 | 21.47 | 28.52 | 25.80 |
| Ties Merging | 37.67 | 33.23 | 39.11 | 28.02 | 21.53 | 28.74 | 31.40 |
| DARE + Ties Merging | 48.17 | 33.56 | 40.40 | 38.83 | 23.81 | 30.18 | 24.43 |
| Consensus Merging | 45.15 | 32.54 | 43.03 | 36.24 | 24.21 | 38.82 | 25.03 |
| PCB Merging | 52.77 | **41.96** | 40.90 | 46.34 | 24.56 | 40.15 | 26.57 |
| EA | 15.92 | 25.06 | 28.23 | 17.36 | 19.29 | 13.31 | 14.48 |
| HM3 | **53.20** | 41.12 | **45.92** | **46.82** | **29.13** | **43.93** | **33.29** |

## C.1.4   Code Generation Task

In the code generation domain, we evaluate on MBPP and HumanEval. MBPP [5] contains 974 beginner-level tasks targeting short Python program synthesis from natural-language prompts; performance is measured with pass@1. HumanEval [9] is a benchmark dataset proposed by OpenAI, specifically designed to evaluate code generation capabilities. The dataset comprises 164 programming problems, where each problem requires the model to generate a Python function based on a natural language description. The evaluation metric of HumanEval is pass@100. The model is allowed to generate up to 100 code solutions for each problem. This metric assesses whether at least one of these generated solutions passes all test cases.

## C.2   Detail of Main Results

**Discussion**   An interesting observation from Table 1 and Table 2 is that the models produced by HM3 sometimes outperform or closely match task-specific fine-tuned models, which are typically considered performance upper bounds for their respective tasks. We attribute this to a key distinction: while fine-tuned models adapt only at the parameter level, HM3 performs both parameter-level and architecture-level merging. As discussed in the introduction, models with different architectures exhibit diverse representational capacities and task preferences, which can extend the performance boundary beyond that of a single fixed architecture. For example, if the layers of the merged model increase, then according to the scaling law, the merged model is expected to have a higher theoretical performance ceiling.

**Merging LLAMA-2-13B model**   Table 5 presents the performance comparison of various merging methods on the LLaMA-15B series across general, translation, math, and code tasks. The proposed HM3 framework achieves consistent and significant improvements over existing classical and SOTA parameter-level and architecture-level baselines. HM3 attains the highest average performance across all task categories, particularly excelling on GSM8K, Xnli, and HumanEval.

**Merging ViT-B/32 model**   As shown in Table 3, the HM3 method consistently outperforms other approaches on ViT-B/32, achieving an average accuracy of 66.91%, which represents a significant improvement. Specifically, HM3 achieved 77.21% and 77.62% on the EuroSAT and SVHN datasets, respectively, and recorded a 68.21% accuracy on the GTSRB dataset, surpassing other methods. Although its performance on the DTD dataset is slightly lower than that of the other tasks, it still outperforms the Ties and Task Arithmetic methods.

**Merging ViT-L/14 model**   Table 4 summarizes the performance of different merging methods on the ViT-L/14 model across various vision tasks. The results indicate that the HM3 method consistently achieved the best performance across most tasks, with particularly high accuracy on the SVHN and GTSRB datasets, reaching 90.48% and 83.43%, respectively. Notably, HM3 achieved an overall average accuracy of 80.30% across all datasets, significantly outperforming the other merging methods.

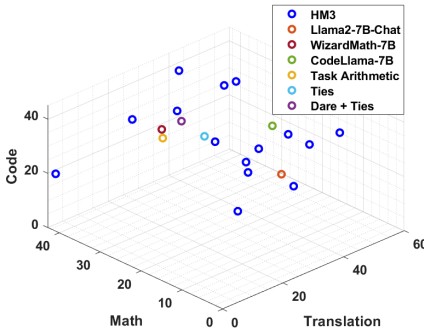

Figure 4: The illustration of different model merging methods in the text translation, math reasoning, and code generation tasks.

## C.3 Detail Analysis of Multi-Objective Model Merging

HM3 is capable of generating a set of approximately Pareto-optimal merged models, which enables adaptation to different user preferences. Unlike other merging methods that produce only a single solution, HM3 yields a diverse set of candidate models. To compare their quality fairly, we compute the Pareto dominance relations by pooling all solutions from different methods together. A solution $x_a$ is said to be dominated by another solution $x_b$ if $x_b$ is no worse in all objectives and strictly better in at least one. The detailed computation procedure is provided in Appendix A.2. Based on this analysis, we evaluate the dominance relations across all solution sets, as shown in Figure 2. Yellow cells indicate that the solution on the vertical axis is Pareto dominated by the one on the horizontal axis, while blue cells denote no dominance. The results show that every competing method is dominated by at least one solution from the HM3 solution set (i.e., S1–S15). To compare with EA, we extend EA to a multi-objective version (MOEA) as a baseline. Since MOEAs also produce a set of solutions, we employ the hypervolume (HV) metric [25, 56], which measures both convergence and diversity. A higher HV value indicates a better overall solution set. HM3 achieves an HV of 1.6824, significantly outperforming the MOEA baseline, which obtains an HV of only 1.1329. This gap highlights the inefficiency of unguided EA-based methods in complex multi-objective spaces.

## C.4 Effect of Different Number of Tasks

In this subsection, we demonstrate the effectiveness of HM3 across different numbers of tasks. In the main text, we illustrated the effectiveness of HM3 on three tasks, and the illustration of metrics for different merging methods is shown in Figure. 4. From it, our approach was capable of producing a set of Pareto-optimal merged models, along with their corresponding metrics, which provided valuable guidance for users to personalize their selection based on the specific needs of their tasks. In contrast, other methods were limited to generating only a single solution.

We provide evidence of HM3's effectiveness on two tasks: code generation and mathematical reasoning. The experimental results are presented in Figure 5 (Figure 3 in the main manuscript), which clearly demonstrate the significant advantages of HM3. Specifically, HM3 is capable of generating a Pareto optimal set of solutions that excel not only in parameter optimization but also in architectural configurations. The blue circles in the figure represent HM3, showing that its solutions are well-distributed across the entire performance curve. Based on the available data, we used convex hull software and Gaussian process fitting to approximate the Pareto front. The results indicate that the solutions produced by HM3 closely approximate a comprehensive Pareto front, effectively capturing the optimal trade-offs under varying conditions. In contrast, other methods, such as Task Arithmetic, Ties, and DARE Ties, are restricted to generating a single optimal solution exclusively in the parameter space, as indicated by the red squares, yellow stars, and green diamonds, respectively. It is evident that the solution sets produced by these methods are confined to narrower regions of the performance spectrum, lacking the diversity and flexibility that HM3 provides. Moreover, these single solutions are noticeably farther from the approximated Pareto front.

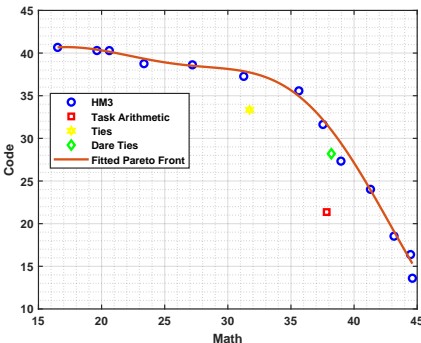

Figure 5: The illustration of different model merging methods in the math reasoning and code generation tasks.

Table 6: Performance of HM3 in different spaces

| Instance | Single Objective | | | Multi-Objective |
| --- | --- | --- | --- | --- |
| | Translation | Math | Code | HV |
| HM3 w.o. para. opt. | 32.21 | 18.36 | 20.67 | 1.3506 |
| HM3 w.o. archi. opt. | 34.01 | 38.51 | 28.67 | 1.6387 |
| HM3 | **44.68** | **45.62** | **43.62** | **1.8120** |

## C.5  Detailed Analysis of Ablation Study

To assess the effectiveness of HM3 in jointly optimizing both the parameter and architecture spaces, we conduct two ablation experiments comparing the following three variants: (i) HM3, (ii) HM3 w.o. arch (without architecture-level optimization), and (iii) HM3 w.o. para (without parameter-level optimization). The results are summarized in Table 6. In the first experiment, we evaluate performance under a single-objective setting using a sampled preference vector. HM3 consistently outperforms the two ablated versions across all tasks, with especially notable gains in code generation. This suggests that the joint optimization of both spaces yields significant synergistic benefits and that architecture-level adaptation plays a crucial role in tasks with more structural complexity. The second experiment examines the HV of these methods in a multi-objective setting. HM3 achieves the highest HV score, followed by HM3 w.o. arch, while HM3 w.o. para performs the worst. These results highlight the importance of parameter optimization in ensuring competitive solutions across objectives, while also demonstrating that architecture optimization further enhances the solution performance.

## C.6  Computational Cost Analysis

Compared to the conventional pretraining and fine-tuning paradigm, HM3 requires significantly less computational power and time to achieve a high-performance model with a novel architecture. Moreover, it eliminates the need for high-quality data for pretraining or fine-tuning. Specifically, the network scale of policy and value networks with 200MB parameters in HM3 is much smaller than the LLM with 13.5GB parameters (i.e., Llama-2-7B) in the fine-tuning stage. In this manner, the computational power required for RL training is significantly lower than for fine-tuning. In this paper, we used A6000 or 3090 GPUs for RL training of HM3, whereas full fine-tuning typically requires A100-level GPU clusters. Additionally, the overall computation time of HM3 is significantly lower than that of fine-tuning. During RL training, only inference (i.e., forward propagation) is required, whereas full fine-tuning necessitates both forward and backward propagation of the LLM. According to time complexity analysis, the backward propagation is typically several times more consuming than the forward. Therefore, the time required for the full fine-tuning is several times greater than that for RL on the same dataset. Finally, we compare the performance of HM3 and the full fine-tuning at a similar time cost, and the results are provided in Table. 7. We can observe that HM3 still obtains the best performance compared with traditional fine-tuning methods at a similar time cost. Given a preference vector, we compared HM3 not only against traditional efficiency baselines such as Full

Table 7: Comparison between HM3 and full fine-tuning methods

| Method | | Translation | Math | Code |
|---|---|---|---|---|
| Fine-Tuning Method | LLAMA-2-7B-Chat | 42.23 | 24.15 | 21.66 |
| | WizardMath-7B | 37.72 | 45.60 | 22.74 |
| | CodeLLama-7B | 36.12 | 18.37 | 43.11 |
| Grid Search | | 29.31 | 21.21 | 23.12 |
| Evolutionary Search | | 31.92 | 23.49 | 26.90 |
| HM3 | | **44.68** | **45.62** | **43.62** |

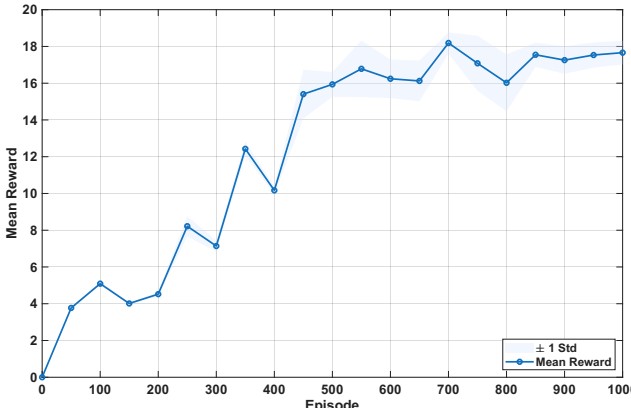

Figure 6: The convergence of RL in the HM3 at the architecture space.

Fine-tuning, but also introduced two additional search-based architecture merging baselines for a more comprehensive efficiency comparison: (i) Grid Search: Under the same maximum inference path length $T_{\max}$ as HM3, this method exhaustively enumerates all possible path combinations until reaching approximately the same number of evaluations as HM3. Layer-wise composition is performed using the passthrough strategy implemented in Mergekit. (ii) Evolutionary Search: The individual encoding shares the same maximum path length as HM3. The search operators follow the standard Differential Evolution algorithm with default hyperparameters and population size. The number of evaluations is aligned with that of HM3. Layers are composed using Mergekit's passthrough method. This approach differs from the EA baseline in the original manuscript, which restricts the search space by controlling model sequences via tokens—a design that significantly reduces complexity but at the cost of final merged model performance. The final performance comparison is summarized in the table. These results demonstrate that, under the same computational budget, HM3 achieves superior multi-task performance compared to other architecture merging methods based on search algorithms.

## C.7 Effectiveness of RL

In this subsection, we delve into the convergence of HM3. Specifically, we randomly sample a preference vector and observe the obtained reward when merging models on text translation, mathematical reasoning, and code generation tasks. The experimental results are illustrated in Figure 6. As shown in Figure 6, the overall reward increases progressively as the number of training episodes increases. During the first 200 episodes, the growth in reward was relatively slow, which is attributed to PPO's exploration phase, where HM3 had not yet accumulated sufficient experience and the policy network had not been trained. However, after episode 200, with the introduction of the experience replay mechanism, the reward begins to rise significantly, indicating that the algorithm is gradually learning from past experiences and improving its policy. As training continues, the reward shows a more stable upward trend and eventually converges to a value close to 18 around the 1000th episode. HM3 can effectively leverage past experiences to optimize its policy and achieve convergence.

# D    Discussions and Limitations

In this section, we discuss some concerns about this work from the multi-objective, architecture-level merging, and future work perspectives.

As for the multi-objective perspective, we set the number of objectives to three, namely translation, math, and code. These objectives are intentionally chosen for their conflicts and diversity, which help produce a well-separated and sparse Pareto front in multi-objective optimization. Adding more objectives that do not bring sufficiently greater diversity, such as a general language understanding objective, would shift the setting into the many-objective regime. That shift increases front dimensionality, densifies the set of solutions, and weakens dominance relations, all of which are known to degrade the effectiveness of standard multiobjective methods. For these reasons, we do not expand the objective set here. If expansion is necessary, pair it with objective selection and decorrelation, decomposition, or reference-vector methods, indicator-based selection, and dimensionality reduction. Future work focuses on many-objective algorithms and high-dimensional discrimination, as well as theoretical foundations for cross-architecture merging and computationally efficient evaluation.

From the architecture-level merging perspective, HM3 is search-based and achieves high efficiency, though search itself can be unstable, sometimes falling into local optima or producing large variance. Compared with parameter-level methods, architecture-level merging typically incurs a higher time cost, a common limitation of current techniques. HM3 performs strongly when merging models that share the same base architecture but target different tasks, extending the performance boundary beyond parameter merging restricted to identical structures. This work is an initial step toward architecture-level merging, focusing on handling post-merge structural differences. This idea parallels SOLAR 10.7B, which concatenates the first twenty and last twenty layers of Mixtral-7B with continued pretraining, while we pursue similar architecture expansion via training-free model merging. When heterogeneous architectures are placed in a single candidate pool, for example, by concatenating layers from Qwen and LLaMA, it becomes nearly impossible to merge them into an effective model. Additionally, existing alignment partially addresses low-order statistics, and deeper semantic shifts remain. Without alignment under cross-architecture settings, the merged model fails on the target tasks. In summary, merging fully heterogeneous architectures is a promising yet unsolved endeavor.

In the future, we plan to focus on several theoretical directions for architecture-level model merging: (i) investigating nonlinear or piecewise mode connectivity under structural variations to reveal the reachability and transition paths in parameter space; (ii) analyzing how mechanisms such as glue layers guarantee intermediate representation consistency, based on representation alignment and information bottleneck theories; and (iii) quantifying the effects of structural merging on generalization error and model expressiveness.

