# OpenReview forum: "HM3: Hierarchical Multi-Objective Model Merging for Pretrained Models"
_NeurIPS.cc/2025/Conference — NeurIPS 2025 spotlight_

### Official Review · Reviewer_wkP4 · 2025-06-11

**Clarity:** 3
**Significance:** 2
**Originality:** 3
**Rating:** 4
**Confidence:** 3

**Summary:**

This work introduces a novel formulation of model merging as a bilevel multi-objective optimization problem that simultaneously explores the parameter and architecture spaces—overcoming the constraint of previous methods that focus only on parameter merging within identical architectures. The upper level conducts parameter fusion guided by user-defined task preferences, while the lower level performs architecture search by identifying optimal inference paths across layers.

**Questions:**

I have some concerns regarding the computational efficiency of HM3, the limited set of parameter-level baselines, and the stability of the RL-based architecture search (see Weakness part). It would be helpful if the authors could address these points in their response.

**Ethical Concerns:**

["NO or VERY MINOR ethics concerns only"]

**Final Justification:**

The proposed method shares the objectives of conventional model merging but offers novel alternative contributions, justifying the adjusted rating.

**Limitations:**

yes

**Paper Formatting Concerns:**

No formatting concerns

**Quality:**

2

**Strengths And Weaknesses:**

- Strengths
1. The paper introduces a novel bilevel multi-objective optimization framework that simultaneously considers both parameter-level and architecture-level merging. This formulation significantly extends the scope of model merging beyond existing parameter-only approaches.
2. The optimization problem is clearly defined with solid theoretical foundations. The use of a Stackelberg equilibrium-based bilevel formulation, together with preference-driven decomposition, makes the overall methodology logically sound and principled.
3. By approximating the Pareto front across multiple objectives, HM3 allows the user to select merged models based on custom task preferences, providing flexibility and interpretability not present in single-solution methods.

- Weaknesses
1. Its bilevel nature and RL-based architecture search introduce substantial computational overhead. Unlike many model merging methods that emphasize efficiency under limited resources, the paper lacks sufficient evaluation on computational cost. The supplementary material mentions a cost analysis, but it is not comprehensive or benchmarked against strong baselines. Table 6 compares only against full fine-tuning, which may not be the most appropriate reference.
2. The paper compares HM3 against only a small subset of merging baselines (Task Arithmetic, TIES, DARE-TIES), while omitting many recent and competitive methods. This weakens the empirical validity of the claimed superiority in parameter-level performance.
3. Figure 2 suggests that all baselines are dominated by at least one HM3 solution, but it remains unclear whether HM3 produces stable and reproducible solutions across runs, given the stochasticity of the RL-based architecture search. Although the appendix provides some analysis on RL effectiveness, it does not fully address potential convergence instability or variance across seeds.
4. There appears to be an error in Equation (13).
5. The feasibility and semantic consistency of stitching layers across different models via a learned MLP-based projection is an underexplored issue. A deeper analysis on representation alignment or performance degradation from misaligned layers would strengthen the paper.

---

> ### Author Rebuttal · Authors · 2025-07-31
>
> **W1:** Thank you for your attention to the time cost analysis. We acknowledge that the original manuscript lacked sufficient detail regarding computational overhead. This omission was primarily due to our view that architecture-level model merging is fundamentally a different task from most existing parameter-level merging methods. The former involves dynamically altering the inference path via search, which inevitably introduces a larger search space and higher computational complexity. However, we believe the necessity of exploring more optimal model compositions beyond fixed architectures should not be overlooked. Therefore, despite the increased complexity, architecture merging constitutes a meaningful research direction that deserves investigation.
>
> To address this concern, we have included two additional **architecture-level baselines** under the same maximum inference path length T_{\max} for comparison:
>
> - **Grid Search**: Exhaustively enumerates possible paths using Mergekit’s passthrough for layer composition until a similar number of evaluations as HM3 is reached.
> - **Evolutionary Search**: Uses differential evolution with standard settings and Mergekit’s passthrough for layer composition, matching HM3's evaluation budget. This differs from the EA baseline in the original submission, which restricted the search space via token constraints but at the cost of final merged model performance.
>
> The performance comparison is shown below:
>
> | Method              | Translation | Math      | Code      |
> | ------------------- | ----------- | --------- | --------- |
> | Grid Search         | 29.31       | 21.21     | 23.12     |
> | Evolutionary Search | 31.92       | 23.49     | 26.90     |
> | **HM3**             | **44.68**   | **45.62** | **43.62** |
>
> These results confirm that HM3 consistently outperforms other search-based merging methods under equivalent computational budgets.
>
> **W2:** As noted in W1, architecture-level merging differs fundamentally from parameter-level merging. While the latter operates on a fixed model topology by interpolating weights, architecture merging constructs new inference paths by combining layers from different pretrained models.
>
> Our original baselines focused on representative parameter-level methods to highlight the benefits of architecture merging. However, we acknowledge that this may have limited the comprehensiveness of the evaluation.
>
> To address this, we have included two SOTA parameter merging methods—PCB-Merging [1] and Consensus Merging [2]—as baselines, recommended in **Reviewer TWdc**. Results show that HM3 consistently outperforms on language ( **see following additional experiment**) and vision tasks (**please see the result in Reviewer TWdc**), reinforcing the effectiveness of HM3.
>
> [1] PCB-Merging: Parameter Competition Balancing for Model Merging, NeurIPS 2024
> [2] Consensus Merging: Localizing Task Information for Improved Model Merging and Compression, ICML 2024
>
> **Additional experiment**
>
> 1.LLAMA-2-13B
>
> | Methods           | General-Glue | Translation-WMT&ISWT | Translation-XNLI | Math-GSM8K | Math-MathQA | Code-HumanEval | Code-MBPP |
> | ----------------- | ------------ | -------------------- | ---------------- | ---------- | ----------- | -------------- | --------- |
> | Consensus Merging | 45.15        | 32.54                | **44.03**        | 36.24      | 24.21       | 38.82          | 25.03     |
> | PCB Merging       | 52.77        | **41.96**            | 40.90            | 46.34      | 24.56       | 40.15          | 26.57     |
> | **HM3**           | **53.20**    | 41.12                | 43.92            | **46.82**  | **29.13**   | **43.93**      | **33.29** |
>
> 2.LLAMA-2-7B
>
> | Methods           | General-Glue | Translation-WMT&ISWT | Translation-XNLI | Math-GSM8K | Math-MathQA | Code-HumanEval | Code-MBPP |
> | ----------------- | ------------ | -------------------- | ---------------- | ---------- | ----------- | -------------- | --------- |
> | Consensus Merging | 47.52        | 37.97                | 40.74            | 37.95      | 27.99       | 30.15          | 37.93     |
> | PCB Merging       | 49.11        | 39.29                | 33.35            | 39.25      | **28.14**   | 32.53          | 39.05     |
> | **HM3**           | **51.04**    | **41.68**            | **40.24**        | **45.62**  | 28.08       | **43.62**      | **44.62** |
>
> 3.Qwen-2.5-1.5B
>
> | Methods           | General -Glue | Translation-WMT&ISWT | Translation-XNLI | Math-GSM8k | Math-MathQA | Code-HumanEval | Code-MBPP |
> | ----------------- | ------------- | -------------------- | ---------------- | ---------- | ----------- | -------------- | --------- |
> | Consensus Merging | 46.42         | 29.82                | 38.09            | 23.98      | 37.84       | 22.59          | 34.08     |
> | PCB Merging       | 47.25         | 30.05                | 40.23            | 24.29      | 36.90       | 21.87          | 41.33     |
> | **HM3**           | **48.22**     | **32.26**            | **41.03**        | **28.05**  | **40.13**   | **34.31**      | **47.80** |
>
>
>
> **W3:** Thank you for your valuable comments. We address your concerns in two parts:
>
> **Regarding Figure 2:** The displayed Pareto front is not from a single run, but represents the mean performance across 10 independent runs, each using a different random seed for each preference vector. For each vector, we generate one merged model per run, evaluate it on the three tasks, and report the average performance over 10 runs.  As a result, we obtain 15 averaged performance points corresponding to the 15 preference vectors (S1–S15). All baselines are also run with 10 different seeds for fair comparison. We will clarify this setup in the revised version.
>
> **Regarding Figure 6:** We agree that the current figure lacks a depiction of variability. To improve it, we re-ran HM3 under the same preference vector for 10 independent trials. The standard deviation of final rewards is consistently below 0.6, confirming the robustness and reproducibility of our method. The updated figure in the revision will show the average reward curve with shaded confidence intervals across episodes to better illustrate convergence stability.
>
> | Episode | Mean Reward (± Std) |
> | ------- | ------------------- |
> | 0       | 0.000 (± 0.000)     |
> | 50      | 3.772 (± 0.049)     |
> | 100     | 5.093 (± 0.099)     |
> | 150     | 4.014 (± 0.046)     |
> | 200     | 4.521 (± 0.120)     |
> | 250     | 8.218 (± 0.558)     |
> | 300     | 7.140 (± 0.300)     |
> | 350     | 12.427 (± 0.391)    |
> | 400     | 10.169 (± 0.204)    |
> | 450     | 15.404 (± 1.323)    |
> | 500     | 15.936 (± 0.675)    |
> | 550     | 16.783 (± 1.539)    |
> | 600     | 16.242 (± 1.047)    |
> | 650     | 16.123 (± 1.111)    |
> | 700     | 18.189 (± 0.572)    |
> | 750     | 17.083 (± 1.497)    |
> | 800     | 16.022 (± 1.547)    |
> | 850     | 17.547 (± 0.657)    |
> | 900     | 17.256 (± 0.745)    |
> | 950     | 17.533 (± 0.683)    |
> | 1000    | 17.662 (± 0.644)    |
>
> **W4:**  Thank you for your constructive feedback on the value function formulation. While Eq. (13) is mathematically correct, we agree its original form may cause ambiguity.
>
> To improve clarity, we revised it as:
>
> $L^{\text{VF}}(\phi) = E_t \[(V\phi(S_t)-\hat{V_t})^2\], \quad \hat{V_t} = V_\phi(S_t) + \hat{A_t}$
>
>
> where $\hat A_t$  is the advantage estimated via generalized advantage estimation. This formulation clearly separates the value prediction $V_\phi(S_t)$ from the target $\hat V_t$, avoiding confusion with expressions like $V_\phi(S_t)-\hat A_t$.
>
> We also include a detailed explanation of GAE in the appendix to support reproducibility and clarity.
>
> **W5:** Thank you for your thoughtful suggestions regarding the feasibility and semantic consistency of cross-architecture layer stitching. We fully agree that maintaining representational alignment is critical when merging layers from heterogeneous models. Our MLP-based projection mechanism is specifically designed to address both dimensional and distributional mismatches.
>
> We revised the manuscript to analyze this issue from three perspectives:
>
> 1. Theoretical and empirical motivation: We explicitly discuss challenges such as hidden size mismatches, differing attention head numbers, normalization statistics, and semantic drift, which are common even across structurally similar models. Prior work [3] also shows that naive layer stitching (e.g., CodeLLaMA vs. LLaMA-2-Chat) significantly degrades performance. Our appendix includes a moment-matching analysis to support how the projection matrix helps align feature distributions.
>
> 2. Empirical validation: We conducted ablation experiments using LLaMA-2-7B to assess the impact of omitting MLP alignment:
>
> | Model                     | Translation | Math      | Code      |
> | ------------------------- | ----------- | --------- | --------- |
> | HM3 without MLP Alignment | 38.15       | 39.83     | 38.21     |
> | **HM3**                   | **44.68**   | **45.62** | **43.62** |
>
> These results confirm that the projection mechanism substantially improves cross-layer semantic alignment and task performance.
>
> 3. Future directions: A new *Discussion* section in the revised manuscript highlights this issue and frames our work as an initial step toward principled cross-model layer composition. We hope it motivates further exploration of lightweight and theoretically grounded alignment strategies.
>
> [3] Y. He, et al. MergeBench: A Benchmark for Merging Domain-Specialized LLMs. arXiv preprint arXiv:2505.10833, 2025.

---

> > ### Comment · Reviewer_wkP4 · 2025-08-05
> >
> > Thank you for the authors’ detailed responses. Although they argue that proposed method is fundamentally different from conventional model-merging techniques, I still consider it part of the model-merging domain given its objectives. The additional “representation alignment” analysis seems to be a simple ablation study, perhaps limited by time. Overall, I feel my main concerns remain unaddressed. Nevertheless, the paper does offer novel contributions as an alternative to traditional model-merging approaches, so I have adjusted my ratings accordingly.

---

> > > ### Author Response · Authors · 2025-08-05
> > >
> > > We sincerely thank the reviewer for the careful reading and valuable feedback. We appreciate your recognition of the novel contributions of our work as an alternative approach within the model merging domain.
> > >
> > > We acknowledge that the current “representation alignment” analysis may appear limited, and more comprehensive ablation studies would further strengthen the work. Due to time constraints, our experiments in the rebuttal period are not yet exhaustive, but we are committed to incorporating additional experiments and deeper analyses in the revised version to address your concerns as fully as possible.
> > >
> > > Thank you again for your valuable suggestions, which are very helpful to improve our paper.

---

### Official Review · Reviewer_TWdc · 2025-07-03

**Clarity:** 3
**Significance:** 3
**Originality:** 4
**Rating:** 5
**Confidence:** 5

**Summary:**

This paper proposes HM3, a hierarchical model merging framework that jointly optimizes both the parameter space and the architecture space. To address the complexity of the combined search space, the authors formulate the problem as a bilevel optimization task, incorporating actor-critic reinforcement learning with policy discretization. Additionally, a multi-objective optimization setup is introduced to support user preferences and task-specific trade-offs. Experiments demonstrate the framework’s ability to generate diverse, reusable, and high-performing merged models under limited resources.

**Questions:**

1, How does HM3 perform when merging models with significantly different architectures or diverse tasks?

2, HM3 retrains and generates different merged models for each task preference vector. Have the authors evaluated whether the learned strategies (e.g., actor-critic policies) can generalize to unseen tasks or new preference vectors without re-optimization from scratch?

3, Since HM3 performs merging across both parameter and architecture spaces, the resulting models and inference paths may vary significantly. Have the authors analyzed the consistency and performance variance of the merged models across multiple runs? Does the stochastic nature of the search lead to unstable final results?

**Ethical Concerns:**

["NO or VERY MINOR ethics concerns only"]

**Final Justification:**

The quality of this paper is improved and I had misunderstand some key contributions of this work before. The idea about bilevel optimization is quite novel.

**Limitations:**

1, No direct comparison with recent SOTA merging baselines weakens empirical validation.

2, The role of hyperparameter tuning is not thoroughly addressed despite its importance in merging.

3, The evaluation scope is narrow, focusing on limited model types and task domains.

**Paper Formatting Concerns:**

This paper is well-writen and no formatting concerns.

**Quality:**

4

**Strengths And Weaknesses:**

Strengths:

1, Innovative idea of merging models across both parameter and architecture spaces.

2, Clear formulation of the problem using bilevel optimization.

3, Use of reinforcement learning improves efficiency in the large search space.

Weaknesses:

1, Lacks comparison with recent state-of-the-art model merging methods (e.g., PCB-Merging, CAT Merging), which limits the empirical strength of the work.

2, Hyperparameter optimization is under-discussed; related works such as Model Evolver and GENOME are not analyzed.

3, Experiments are conducted on relatively narrow model and task settings, which may reduce the generalizability of the findings.

Reference: \
1), ACL24, Model Evolver: Knowledge Fusion By Evolving Weights of Language Models \
2), NeurIPS24, PCB-Merging: Parameter Competition Balancing for Model Merging \
3), ICML25 CAT Merging: A Training-Free Approach for Resolving Conflicts in Model Merging \
4), GENOME, Nature-Inspired Population-Based Evolution of Large Language Models

---

> ### Author Rebuttal · Authors · 2025-07-31
>
> **W1:** We thank the reviewer for the insightful suggestion. While our original focus was on classical baselines to emphasize HM3's joint parameter- and architecture-space merging capability, we acknowledge the importance of including recent SOTA methods. Accordingly, we have made the following updates:
>
> 1. Inclusion of PCB Merging: We re-implemented PCB Merging and evaluated it across all tasks, including text and vision. The results are included in the revised manuscript.
>
> 2. Consensus Merging as a substitute for CAT Merging: As CAT Merging was recently accepted to ICML 2025 and lacks public source code, we are working to reimplement it. Meanwhile, we added Consensus Merging [1], a strong baseline emphasized in the CAT Merging paper.
>
> 3. Empirical insights: PCB Merging initially outperformed the original HM3 due to its high-quality merged model. After incorporating PCB-generated models into HM3's candidate pool, HM3 significantly outperformed PCB, highlighting its flexibility and strength in architecture-level merging.
>
> 4. Updated related work: We revised the *Related Work* section to discuss PCB, CAT, and Consensus Merging, offering a more comprehensive literature review.
>
>    [1] Wang et al., Localizing Task Information for Improved Model Merging and Compression, *ICML*, 2024.
>
> **Additional experiment**
>
> 1. ViT-B/32
>
> | Methods           | SUN397 | RESISC45 | SVHN  | GTSRB | DTD   | MNIST | Cars  | EuroSAT | Average   |
> | ----------------- | ------ | -------- | ----- | ----- | ----- | ----- | ----- | ------- | --------- |
> | Consensus Merging | 64.73  | 73.51    | 79.46 | 69.03 | 52.63 | 96.89 | 63.06 | 77.20   | 72.06     |
> | PCB Merging       | 63.58  | 75.47    | 82.31 | 72.57 | 54.78 | 97.12 | 64.42 | 79.63   | 73.74     |
> | Original HM3      | 63.22  | 73.27    | 77.62 | 68.21 | 52.22 | 96.94 | 64.02 | 78.06   | 71.70     |
> | **HM3**           | 63.42  | 76.27    | 82.11 | 73.11 | 54.60 | 96.85 | 64.63 | 79.66   | **73.83** |
>
> 2. Vit-L/14
>
> | Methods           | SUN397 | RESISC45 | SVHN  | GTSRB | DTD   | MNIST | Cars  | EuroSAT | Average   |
> | ----------------- | ------ | -------- | ----- | ----- | ----- | ----- | ----- | ------- | --------- |
> | Consensus Merging | 73.39  | 88.05    | 87.43 | 81.16 | 66.04 | 98.88 | 84.26 | 90.81   | 83.75     |
> | PCB Merging       | 75.04  | 88.75    | 86.46 | 86.55 | 69.13 | 98.91 | 86.01 | 91.01   | 85.23     |
> | Original HM3      | 72.85  | 88.00    | 90.48 | 83.43 | 66.72 | 98.41 | 86.17 | 89.50   | 84.45     |
> | **HM3**           | 74.76  | 88.43    | 90.02 | 85.17 | 70.21 | 98.44 | 86.33 | 89.32   | **85.34** |
>
> 3.LLAMA-2-13B
>
> | Methods           | General-Glue | Translation-WMT&ISWT | Translation-XNLI | Math-GSM8K | Math-MathQA | Code-HumanEval | Code-MBPP |
> | ----------------- | ------------ | -------------------- | ---------------- | ---------- | ----------- | -------------- | --------- |
> | Consensus Merging | 45.15        | 32.54                | **44.03**        | 36.24      | 24.21       | 38.82          | 25.03     |
> | PCB Merging       | 52.77        | **41.96**            | 40.90            | 46.34      | 24.56       | 40.15          | 26.57     |
> | **HM3**           | **53.20**    | 41.12                | 43.92            | **46.82**  | **29.13**   | **43.93**      | **33.29** |
>
> 4.LLAMA-2-7B
>
> | Methods           | General-Glue | Translation-XNLI | Math-MathQA | Code-MBPP |
> | ----------------- | ------------ | ---------------- | ----------- | --------- |
> | Consensus Merging | 47.52        | 40.74            | 27.99       | 37.93     |
> | PCB Merging       | 49.11        | 33.35            | **28.14**   | 39.05     |
> | **HM3**           | **51.04**    | **40.24**        | 28.08       | **44.62** |
>
> 5.Qwen-2.5-1.5B
>
> | Methods           | General-Glue | Translation-XNLI | Math-MathQA | Code-MBPP |
> | ----------------- | ------------ | ---------------- | ----------- | --------- |
> | Consensus Merging | 46.42        | 38.09            | 37.84       | 34.08     |
> | PCB Merging       | 47.25        | 40.23            | 36.90       | 41.33     |
> | **HM3**           | **48.22**    | **41.03**        | **40.13**   | **47.80** |
>
>
>
> **W2:**  We thank the reviewer for the valuable feedback. In original version, hyperparameters were tuned via grid search on smaller models and then transferred to larger ones. The revised manuscript includes further clarifications:
>
> - Expanded hyperparameter details: We now list key hyperparameters (e.g., RL learning rate, Wolpertinger neighborhood size, number of subproblems) with value ranges. These were tuned on Qwen‑1.5B and ViT‑B/32, then applied to LLaMA‑2‑7B and ViT‑L/14. Further experiments are ongoing and results will be provided for the final version.
> - Updated related work: We added *Model Evolver* and *GENOME* to our review. Both use evolutionary method to merge weights in fixed architectures and need to restart for each task.
> - Comparison with HM3: Unlike the above, HM3 incorporates RL for architecture-level search, supports cross-model layer stitching, and adopts multi-objective scalarization for diverse solutions. The learned policy generalizes across tasks and model pools, enhancing reuse and efficiency. We also discuss potential combinations of evolutionary search and RL to further improve joint merging capabilities.
>
> **W3:** We thank the reviewer for emphasizing the importance of generalization. Our original evaluation covered three core LLM task domains, math reasoning, code generation, and multilingual translation, following the Qwen-3 evaluation protocol (e.g., Table 3 in the Qwen-3 technique report). To strengthen generalization, we have expanded both task and model coverage:
>
> Expanded Task domain: Added **GLUE benchmark** for **generative task domain**.
>
> Expanded Tasks:
>
> - *Multilingual (Translation):* Added **XNLI** dataset.
> - *Math:* Added **MathQA** dataset.
> - *Code:* Added **MBPP** dataset.
>
> Expanded Models:
>
> - Added **LLaMA-2-13B** and fine-tuned variants.
>
> These enhancements further demonstrate the robustness and broad applicability of our method across tasks and model sizes.
>
> **Q1:** We thank the reviewer for the insightful feedback and respond from two angles: model compatibility and task generalization.
>
> **Model Compatibility**: HM3 effectively merges models with identical architectures but different specializations. Notably, HM3 also supports architecture-level merges, unlike standard parameter merging, allowing flexible architectural extensions. This capability draws inspiration from SOLAR‑10.7B [2], which significantly improved performance by concatenating the first 20 layers and last 20 layers of Mixtral‑7B followed by continued pretraining. HM3 aims to achieve similar architecture expansion in a training-free manner through model merging, without the need for expensive retraining.
>
> However, merging models with fundamentally different architectures (e.g., Qwen vs. LLaMA) remains difficult. Severe mismatches in dimensions and representations can significantly degrade performance. Our MLP alignment mitigates shallow-level discrepancies but cannot fully bridge deeper semantic gaps. Without alignment, merged models fail even on simple benchmarks.
>
> While HM3 does not yet support fully heterogeneous model merging, it serves as a promising step toward that goal. We will clarify these limitations and discuss future extensions in the revised version.
>
> | Merging LLAMA + Qwen      | Translation | Math     | Code     |
> | ------------------------- | ----------- | -------- | -------- |
> | HM3 without MLP Alignment | 0           | 0        | 0        |
> | **HM3**                   | **11.68**   | **9.21** | **5.13** |
>
> **Task Generalization:** We initially evaluated HM3 on math, code and translation task domains, following Qwen‑3's domain grouping. In the revised version, we expand coverage to include the **General** task domain and an additional task per existing task domain. Results confirm HM3's strong performance across diverse tasks, highlighting its generalization ability.
>  [2] D. Kim et al., *Solar 10.7B: Scaling Large Language Models with Simple Yet Effective Depth Up-Scaling*, arXiv:2312.15166, 2023.
>
> **Q2:** We thank the reviewer for emphasizing the importance of policy generalization. To evaluate reusability, we conducted two new experiments:
>
> - Cross-domain transfer: A policy trained on translation, math, and code tasks was tested on unseen domains (QA, WSD, sentence completion), following the out-of-domain setup in PCB Merging. Results will be added to a new *Out-of-Domain Generalization* subsection.
> - Zero-shot preference generalization: A well-trained policy network in HM3 was applied to two unseen vectors without further search, directly generating inference paths for model merging. The result is the following table.
>
> | Zero-shot preference generalization  | Translation | Math  | Code  |
> | ------------------------------------ | ----------- | ----- | ----- |
> | HM3                                  | 43.86       | 39.05 | 36.85 |
> | Re-train HM3 based on sampled vector | 45.93       | 40.21 | 38.37 |
>
> These experiments evaluate generalization across both task domains and preference vectors. Results show that HM3’s policy generalizes effectively in a zero-shot setting, producing competitive merging paths. We will provide complete results and implementation details in the final revision to highlight the policy’s robustness and reusability.
>
> **Q3:** We thank the reviewer for highlighting the importance of result stability. As HM3 performs joint search over parameter and architecture spaces, we address randomness by running the full pipeline 10 times per preference vector using different seeds. As shown in Figure 6, we report the mean and standard deviation across runs. The final reward is shown **in W3 of Reviewer wkP4**, which demonstrates the stability of HM3.

---

> > ### Comment · Reviewer_TWdc · 2025-08-03
> >
> > Thanks for the response. All my concerns has been addressed and I decide to improve my ratings. The quality of this paper is very good and deserves to be accepted.

---

> > > ### Author Response · Authors · 2025-08-03
> > >
> > > We sincerely appreciate your strong support for our paper and your insightful review comments. Your feedback has been extremely valuable to us, and these comments will be carefully addressed in the revised version.

---

### Official Review · Reviewer_21Mn · 2025-07-03

**Clarity:** 3
**Significance:** 3
**Originality:** 3
**Rating:** 5
**Confidence:** 3

**Summary:**

This paper proposes HM3, a novel hierarchical framework for merging multiple pretrained and fine-tuned models into a single model. The method treats model merging as a multi-objective optimization task and uses reinforcement learning to explore both parameter and architecture space. The result is a Pareto front of merged models, enabling flexible trade-offs among task-specific performance objectives. Experimental evaluations on several language tasks and vision tasks demonstrate promising performance over baseline model-merging techniques.

**Questions:**

1. The authors claim that HM3 supports merging in architecture space. However, all experiments are conducted using pretrained models with highly similar structures. Could the authors provide more details or ablation studies to demonstrate the effectiveness of architecture-level merging, particularly in scenarios involving structurally different components?
2. Did the authors ensure a fair comparison between the MOEA-based baseline and the proposed method? The experimental settings and evaluation are not clearly described. Clarification is needed regarding how performance metrics, computational budgets, and parameter configurations were aligned across methods.
3. How do the authors select a final model from the set of Pareto-optimal solutions in practice? Is there an automatic or heuristic-based selection strategy, or does it require manual validation? Please elaborate on the model selection process and whether it applies fairly across all compared methods.
4. The paper uses (1, 1) as the reference point for hypervolume (HV) calculation. This choice may not be appropriate in practice as the boundary points may not be included in the evaluation. Have the authors considered using a more appropriate reference point, such as a slightly larger value than the estimated nadir point?
5. Is the proposed method scalable with respect to the number of tasks or objectives? For example, how would HM3 perform when merging models fine-tuned on more than three tasks?

**Ethical Concerns:**

["NO or VERY MINOR ethics concerns only"]

**Final Justification:**

The authors have addressed my concerns and improved the clarity of the paper. Therefore, I have decided to raise my score.

**Limitations:**

Yes

**Quality:**

3

**Strengths And Weaknesses:**

Strengths:
- The combination of parameter- and architecture-level merging within a multi-objective optimization framework is novel and well-motivated.
- The reported experimental results appear promising and demonstrate competitive performance compared to existing model merging methods.

Weaknesses:
- The experimental procedure is insufficiently explained.
- The scalability of the proposed method is not well discussed. While the method is tested on a small number of tasks and model pairs, it is unclear how HM3 performs when scaling to more base models, larger architectures, or more objectives.
- The paper contains typos (e.g., "optimity") and inconsistent symbol usage (e.g., in Appendix A.2.1, the symbols k, m, and d are used ambiguously for the number of objectives). In addition, the language can be improved throughout the paper to eliminate grammatical issues and enhance readability.

---

> ### Author Rebuttal · Authors · 2025-07-31
>
> **W1:** We thank the reviewer for the valuable suggestions. Due to space constraints,  we provided a concise overview of the models and tasks used in the evaluation.  In the revised version, we have addressed the concern about insufficient experimental details with the following clarifications and additions:
>
> **Dataset splits and evaluation protocol:** Each task adopts a 70/30 train/test split. The RL search strictly uses training data, and final performance is evaluated on the held-out test set to avoid data leakage. Task-specific metrics are now described in full, including their computation procedures.
>
> **Hyperparameter settings and stability:** We have added a dedicated *Hyperparameter List* subsection in the appendix:
>
> - Maximum number of iterations: 1000
> - Policy and value networks begin updating after iteration $iter_0 = 200$
> - PPO clipping ratio: 0.1
> - Discount factor $\gamma$: 0.990
> - GAE coefficient $\beta_A$: 0.95
> - Loss coefficients: $c_1 = 1.0$ (value loss), $c_2 = 0.15$ (policy entropy)
>
> We assess robustness via 10 runs with different seeds, reporting mean and standard deviation of rewards to illustrate training stability (**see the result in W3 of Reviewer wkP4**).
>
> **Runtime Environment:** In the *Experimental Platform* section, we now report detailed hardware and software configurations. Qwen‑2.5‑1.5B was evaluated on four 3090 GPUs (24GB each), while LLaMA‑2‑7B and LLaMA‑2‑13B were evaluated on four A6000 GPUs (48GB each). All models can also be deployed on a single GPU. The software versions, dependency libraries, and parallelism settings are listed in the appendix.
>
> To support reproducibility, we will release the source code upon paper acceptance.
>
>
>
> **W2/Q5:** We thank the reviewer for the insightful comments on the scalability of HM3. This paper primarily focuses on architecture-level model merging and demonstrates the superiority of our approach over parameter-based baselines on several representative tasks and model scales. In response to the reviewer’s concerns, we provide a comprehensive discussion from three perspectives, i.e., **tasks**, **models**, and **objectives**, and supplement our claims with additional experiments.
>
> **Task-Level Scalability**: Beyond Translation, Math, and Code, we incorporate **GLUE benchmark** as a new task domain, **Generative/General Task** . Additionally, we further expand task diversity in the current task domains:
>
> - Translation: **XNLI**
> - Math: **MathQA**
> - Code: **MBPP**
>
> **The results are provided in W2 of Reviewer wkP4**.  HM3 maintains superior performance across all four domains, demonstrating robustness to both task number and variety.
>
> **Model-Level Scalability**: HM3 relies on layer-wise features for trajectory selection, ensuring stable complexity regardless of model pool size. We also scale to **LLaMA‑2‑13B** and observe consistent gains, confirming effectiveness on larger models. **The results of LLaMA‑2‑13B are provided in W2 of Reviewer wkP4 or W1 of  Reviewer TWdc**.
>
> **Objective-Level Scalability**: In this work, we set the number of objectives to three—translation, math, and code—chosen due to their inherent conflicts and diversity, which are essential for forming a well-separated and sparse Pareto front in multi-objective optimization. Adding more objectives without sufficiently higher diversity (e.g., general language understanding) would shift the problem into the many-objective regime. This introduces challenges such as excessive front dimensionality, dense solution distributions, and dominance relation degradation, which are known to undermine the effectiveness of traditional multi-objective methods [1].
> For these reasons, we do not include more objectives. But we will explicitly discuss the impact of objective count on problem complexity in the *Discussion* and *Limitations* sections, and outline future directions such as designing dedicated algorithms for many-objective model merging.
>
> [1] B. Li, et al. Many-objective evolutionary algorithms: A survey. *ACM Computing Surveys*, vol. 48, no.1, pp. 1-35, 2015.
>
>
>
> **W3:**  We thank the reviewer for pointing out the spelling and notation issues. We have addressed them as follows:
>
> **Spelling**: All typos (e.g., “optimity” → “optimality”) have been corrected through a full proofreading pass.
>
> **Notation consistency:** We have standardized the notation throughout the paper as follows:
>
> - Number of objectives: $K$
> - Dimensionality of decision space: $d$
> - Layer index: $l$
> - Model index: $m$
>   All notations are now consistently used and clearly defined upon their first introduction in the main text.
>
> **Appendix:** Section A.2.1 has been revised for consistent indexing, and a *Notation Summary Table* has been added for clarity.
>
> We believe these revisions improve both clarity and formal consistency.
>
>
>
> **Q1:**  We sincerely thank the reviewer for their thoughtful comments on the concept of *architecture-space merging*. We would first like to clarify that the core innovation of our method lies in its ability to not only merge model parameters but also to extend or reorganize the architecture of the resulting model. This architectural flexibility offers the potential to overcome performance bottlenecks of a fixed architecture by enhancing the model’s representational capacity and task adaptability.
>
> According to the *scaling laws*, such increases in more parameterized layers can improve model performance. For instance, SOLAR-10.7B [2] improved performance by concatenating the first 20 layers and the last 20 layers from Mixtral-7B, followed by continued pretraining. Our goal is to achieve similar architectural expansion via a training-free model merging.
>
> In this work, we focus on architecture-level merging models via base models with similar architectures due to:
>
> - Functional alignment, preserving semantic consistency after stitching.
> - Reduced distributional shift, as models from the same base checkpoint are more compatible under MLP-based projection.
> - Practicality, since merging fully-heterogeneous architectures (e.g., Qwen with LLaMA) leads to non-functional models due to severe misalignment (**see the result in Q1 of Reviewer TWdc**).
>
> Thus, HM3 represents an intermediate step between traditional parameter merging and full architectural merging, similar to SOLAR-10.7B, but without additional training.
>
> We will clarify these constraints and future directions in the *Limitations* and *Future Work* sections.
>
> [2] D. Kim, et al. Solar 10.7 b: Scaling large language models with simple yet effective depth up-scaling. *arXiv preprint arXiv:2312.15166*, 2023.
>
>
>
> **Q2:**  We thank the reviewer for the suggestion. We agree that including an MOEA baseline is important for fair comparison. While results were briefly reported (**Line 309**), we now provide further clarification.
>
> **Experimental setup:** Both HM3 and the MOEA baseline adopt the same decomposition-based multi-objective framework, sharing:
>
> - Scalarization strategy
> - Initialization seeds (10 per preference vector)
> - 10 independent runs per method, with averaged results
>
> **Implementation:** The MOEA baseline uses decomposed-based MOEA with differential evolution operator (pop. size = 30, crossover = 0.9, mutation = 0.1). Merging is performed via mergekit passthrough, aligned with evolutionary search method **in W1 of Reviewer wkP4**. Both methods use 4 GPUs under comparable evaluation budgets.
>
> **Results:** Using a reference point (1.1, 1.1, 1.1), HM3 consistently achieves higher hypervolume than MOEA on LLAMA-2-7B across all runs, demonstrating superior search efficiency.
>
> | Method  | HV (ref = (1.1, 1.1, 1.1)) |
> | ------- | -------------------------- |
> | MOEA    | 1.2133                     |
> | **HM3** | **1.8120**                     |
>
>
>
> **Q3:**  We thank the reviewer for the important question on preference specification. Our framework assumes users typically do not provide explicit preferences and supports two practical modes:
>
> 1. Offline preference sampling (Default): We uniformly sample preference vectors to approximate the Pareto front. Users can later select a model matching their needs—no input required.
> 2. Optional user preference injection: If a user specifies a preference (e.g., prioritizing translation), we select the closest model on the front or conduct a targeted search, enabling both automated and interactive use.
>
> To ensure a fair comparison protocol, we sample the same preference vector across all methods when comparing HM3 with single-objective baselines, ensuring consistent optimization and evaluation conditions.
>
> We will add a *“Deployment Considerations”* subsection in the appendix and clarify in the main text that all baselines are evaluated under the same preference settings.
>
>
> **Q4:**  We thank the reviewer for the insightful comment on HV calculation. We acknowledge the initial reference point was suboptimal and have corrected it during the review phase.
>
> We now use (1.1, 1.1) for bi-objective cases, and (1.1, 1.1, 1.1) for tri-objective cases.
>
> This follows standard practice in toolkits like PlatEMO [3], where the reference point is set slightly beyond the estimated nadir to ensure valid HV computation and full Pareto coverage.
>
> Experimental results have been updated accordingly, and Line 606 in the manuscript now states:
>
> > “The reference point is set slightly beyond the nadir (e.g., (1.1, 1.1, 1.1)) to ensure non-zero HV and full coverage of the Pareto front.”
>
> We believe this addresses the concern and improves clarity in our evaluation.
>
> | Method               | HV     |
> | -------------------- | ------ |
> | HM3 w.o. para. opt.  | 1.3506 |
> | HM3 w.o. archi. opt. | 1.6387 |
> | **HM3**              | **1.8120** |
>
> [3] Y. Tian, et al. PlatEMO: A MATLAB platform for evolutionary multi-objective optimization. IEEE Computational Intelligence Magazine, vol. 12, no. 4, pp. 73-87, 2017.

---

> > ### Comment · Reviewer_21Mn · 2025-08-06
> >
> > Thank you to the authors for their responses to my questions. I believe the clarity of the paper has improved.

---

> > > ### Author Response · Authors · 2025-08-06
> > >
> > > We sincerely thank you for your thoughtful feedback and positive comments. We are pleased that our responses have helped improve the clarity of the paper. These comments will be carefully addressed in the revised version. Once again, we greatly appreciate your valuable time and constructive suggestions, which have significantly contributed to enhancing our manuscript.

---

### Official Review · Reviewer_B6zw · 2025-07-13

**Clarity:** 3
**Significance:** 3
**Originality:** 3
**Rating:** 5
**Confidence:** 4

**Summary:**

The paper proposes HM3, a method for heterogeneous model merging that extends to both parameters and architecture. It is framed as a hierarchical multi-objective optimization problem.

Specifically, the framework uses a two-level optimization structure to decouple the complex joint search space:

* Parameter Level: It uses existing parameter merging techniques (e.g., DARE-TIES) to efficiently generate a parameter-optimal merged model according to different task preferences, which are defined by multi-objective weight vectors.

* Architecture Level: It models architecture search as a Markov Decision Process (MDP). An Actor-Critic reinforcement learning policy is then designed to search for the optimal inference path (i.e., model architecture) within a "layer pool" composed of layers from the original models.

**Questions:**

In RL, how is the reward calculated for an incomplete network?

The MLP aligns the layer dimensions. What are the specific implementation details for the layer selection?

**Ethical Concerns:**

["NO or VERY MINOR ethics concerns only"]

**Final Justification:**

* The authors successfully clarified that their approach is a novel architecture-level merging, which is fundamentally different from traditional parameter-level merging. This resolved the reviewer's primary confusion about the fusion process.

* They effectively addressed concerns about the introduced MLP layers by explaining that their training is a lightweight process confined to the architecture search phase and that the final merged model requires no subsequent fine-tuning, making it efficient and readily usable.

* The authors' firm commitment to publicly releasing the complete source code upon acceptance assured the reviewer of the work's reproducibility and potential impact on the community.

**Limitations:**

Yes.

**Paper Formatting Concerns:**

No.

**Quality:**

3

**Strengths And Weaknesses:**

Strengths

* The article is well-written.
* It generates a Pareto front, giving users the ability to select the most suitable model based on specific task requirements.
* It utilizes an Actor-Critic framework, the Wolpertinger policy, and reinforcement learning to solve complex path-search problems, which is more guided and efficient than traditional EA.

Weaknesses

* More details on the reinforcement learning aspect are desired.
* The outer-loop optimization in the paper requires the same parameter initialization, which is still confined to the same architectural limitations. Therefore, we cannot refer to it as a hierarchical model fusion.
* The bi-level optimization in the paper is more analogous to alternating optimization (as both levels are overly independent).

---

> ### Author Rebuttal · Authors · 2025-07-31
>
> **W1/Q1:** We thank the reviewer for the suggestion regarding the reinforcement learning details. In the revised manuscript, we have provided the clarifications from the method and experiment aspects:
>
> **Method Aspect**: We first clarify the components of RL. In this work, the architecture-level model merging problem is formulated as an MDP:
>
> 1)State Space: We define the state as the trajectory of selected models and layers. Then, the model index, layer index, and layer parameters are embedded by learnable encoders, and map the trajectory into a state vector via a GRU.
>
> 2）Action Space: The action is defined as selecting the next discrete pair of model index and layer index from the candidate model pool. As a discrete action space, we design a **Wolpertinger discretization strategy** to enhance the efficiency of discrete sampling and assist in selecting the next action.
>
> 3）Reward Function: To encourage efficient inference paths while ensuring performance, the total reward of a trajectory is defined as a weighted performance minus a penalty for the path length.
>
> **Q1**: The reward function is computed after the entire inference path (trajectory) is generated and the MLP-based alignment is performed. The resulting model is evaluated, and the obtained reward is uniformly assigned to all time steps in the trajectory as $R_t$, which is used to train both the policy and value networks. In the original version, the description of the reward function may cause a misunderstanding.
>
> Therefore, in the revised manuscript, we updated the reward formula as:\\[R = \sum_{k=1}^K \lambda_k^{(i)} f_k(\boldsymbol{\theta},~\pmb{h}) - \beta_1 T\\] to indicate that reward obtained by model evaluation after obtaining the complete inference path and performing MLP alignment, thus ensuring the rigor and correctness of reward calculation.
>
> Based on the above MDP, the execution of the RL is the following stages:
>
> **Stage 1: Input and initialization**: The algorithm begins by sampling $N$ preference vectors, each corresponding to a decomposed subproblem in the multi-objective framework. For each preference vector, an existing parameter-level merging method is applied to obtain an initial merged model parameter. Meanwhile, the parameters of the policy, value, and the MLP network for alignment are initialized.
>
> **Stage 2: Trajectory collection and MLP alignment:** For each preference vector, an inner loop is executed in RL. In each iteration, the parameter-level merged model and the fine-tuned models are evaluated to compute the reward and stored in the trajectory buffer. Then, at each step, the current trajectory-encoded state is used to generate a proto-action, which is discretized via the Wolpertinger policy to select the next action. The new state is converted, and the transition is recorded in the buffer. Once a complete path is collected, the algorithm performs MLP alignment.
>
> **Stage 3: Reward computation and network update：** After the alignment, the reward is computed and then is assigned to all time steps in the trajectory. Once the iteration number exceeds $iter_0$, the algorithm enters the network update phase. In this phase, a mini-batch is  sampled from the buffer to compute GAE and target returns. The policy, value network, and MLP alignment network are updated. The entire process continues until *Max_iter* is reached, yielding the final policy network and the optimal inference paths. The well-trained networks can be reused for new tasks or model pools.
>
> **Experiment Aspect: **The hyperparameters of RL are: *Max_iter* is 1000; the networks begin updating after $iter_0 = 200$; the PPO clipping ratio is 0.1;  $\gamma$ is 0.990; the GAE smoothing factor $\beta_A$ is 0.95. The loss coefficients are $c_1 = 1.0$ and $c_2 = 0.15$.
> We split the dataset where 70% is used for RL inference evaluation, while the 30% is reserved for the evaluation of the obtained merged model.
> Finally, in the revised version, we conduct 10 independent runs with different random seeds and report the mean and variance to assess the stability of the RL policy training. The detailed result of RL convergence is provided in **W3 of Reviewer wkP4**.
>
> **W2:** We sincerely thank the reviewer for the valuable feedback and fully understand the underlying concern. While it is true that parameter merging generally requires the *same parameter initialization*, this condition does not apply to the outer-loop optimization in HM3.
>
> To elaborate, in HM3, the outer-loop optimization selects a source model for each layer based on $(m_t, l_t)$, and parameter merging is only performed among models in the pool that share the same architecture as the selected source model. If the selected layer originates from a model that has no architectural counterparts in the pool, parameter merging is simply skipped. Therefore, the outer-loop optimization itself does not rely on a unified parameter initialization across models.
>
> Our approach indeed involves optimization over both model parameters and model architecture, reflecting a two-level decision process. To clarify this point, we have revised the manuscript to provide a more direct explanation of HM3 as follows:
>
> > The lower-level optimization searches for the optimal merged model architecture. The resulting architecture determines the length of the inference path (i.e., the number of layers to be merged). The upper-level optimization then operates on the parameter set $\\{\theta_{m_t,l_t}\\}_{t=1}^T$ corresponding to this architecture.
> >
> > Consequently, the optimal architecture found by the lower level dynamically determines the dimensionality and scale of the parameter search space for the upper level. This naturally forms a hierarchical decision-making structure—first optimizing the model architecture, then optimizing the corresponding model parameters—which embodies the core *hierarchical* nature of our HM3 method.
>
> We hope this clarification resolves the misunderstanding and highlights the distinct roles of architecture and parameter optimization.
>
> **W3:** We greatly appreciate the reviewer’s insightful concern. We understand that the interaction between the upper- and lower-level appears to resemble an alternating optimization. However, we would like to emphasize that HM3 is fundamentally a bilevel optimization rather than an alternating optimization procedure. This distinction is evident in the following aspects:
>
> (i) Different objective functions across levels: The upper-level optimization maximizes the performance $\mathcal{F}$ of the merged model, while the lower-level optimization is formulated as an MDP with a reward function that combines model performance and inference path length: $R = \mathcal{F} - \beta_1 T$. The two levels optimize different functions characterized as bilevel optimization, while alternating optimization decomposes a single objective into interleaved subproblems.
>
> (ii) Strict dependency between levels: the upper-level optimization in HM3 can only be performed *after* the lower-level has searched an optimal architecture. This requirement of solving the lower-level problem to optimality before proceeding to the upper-level is a key feature of bilevel optimization.
>
> (iii) Variable interaction: Interaction variables such as the inference path length $T$ are determined exclusively by the lower-level architecture search and are then passed to the upper-level optimizer.
>
> To avoid confusion, we have revised the manuscript to update $\mathcal{P}2$, making clear the essential distinction between the objectives of the two levels and their interaction structure.
>
> **Q2**: We begin with sampling preference vectors. For each preference vector, the layer selection process is modeled as an MDP, as detailed in W1/Q1. Specifically, the trajectory of selected layers is embedded into a state vector via a learnable encoder. Based on this state, the policy network outputs a continuous proto-action vector, which is mapped to a discrete action using the Wolpertinger policy: it first identifies the nearest neighbors based on the distance to the proto-action vector, constructs a candidate set, and then selects the action that maximizes the value network or performs random exploration. The final action determines the next layer to be appended to the inference path.
>
> This process yields an inference sequence. For every adjacent layer in this sequence, we apply an MLP-based alignment to address potential mismatches in dimension and representation. Specifically, we first detect their input/output dimensions. Then, for each layer, we compute statistical descriptors of its representation, i.e., mean, variance, and distribution characteristics. The proposed MLP network generates a projection matrix that transforms the representation of the preceding layer to align it with the following one in terms of both dimensionality and distribution. The transformed features are then seamlessly passed to the subsequent layer, enabling compatible inter-layer connections across different models. Preliminary experiments indicate that incorporating MLP-based alignment leads to improvements in final performance.
>
> | LLAMA-2-7B                | Translation | Math      | Code      |
> | ------------------------- | ----------- | --------- | --------- |
> | HM3 without MLP Alignment | 38.15       | 39.83     | 38.21     |
> | **HM3**                   | **44.68**   | **45.62** | **43.62** |
>
> MLP-based alignment is applied between adjacent layers in the inference path. As in **W5 of Reviewer wkP4**, learning-based layer stitching via MLP projection remains underexplored. Therefore, in the revised version, we state that the proposed MLP-based projection is in its early exploratory stage. In the *Discussion* section, we outline possible future directions.

---

> ### Author Response · Authors · 2025-08-06
>
> Dear Reviewer B6zw,
>
> We would like to kindly ask whether our responses have addressed your concerns. If there are any remaining questions or points that require further clarification, we would be more than happy to discuss them.
>
> Thank you once again for your valuable time and insightful feedback! We truly appreciate your thoughtful review and constructive suggestions, which have greatly helped us improve our work.

---

> > ### Comment · Reviewer_B6zw · 2025-08-07
> >
> > Thank you for your response and for sharing your innovative method. I'd like to clarify a few points:
> >
> > 1.  Regarding the fusion process, my understanding is that each layer in the final model is selected in its entirety from one of the source models, rather than averaging the parameters of a single layer from multiple models (as is common in parameter-merging techniques). This seems to diverge from some standard model merging approaches. Could you please elaborate on the theoretical basis for this, perhaps something analogous to Linear Mode Connectivity (LMC)?
> >
> > 2.  Due to the introduction of MLP, the fused model isn't immediately usable. Does this necessitate a subsequent training phase? If so, could you provide an estimate of the data requirements and computational overhead involved?
> >
> > 3. Is it like outcome reward?
> >
> > 4.  May I also ask if you plan to release the code, including the details of the reinforcement learning implementation?
> >
> > Thank you again for your time and insights.

---

> > > ### Author Response · Authors · 2025-08-07
> > >
> > > Thank you very much for your response and valuable comments. I will address your points in detail as follows.
> > >
> > > 1. Thanks for your valuable comment. First, we would like to clarify your comment regarding the merging process. In the HM3 framework, the upper-level optimization in the bi-level framework performs parameter-level merging based on the optimal architecture (inference path) obtained from the lower-level optimization. Specifically, for each layer in the optimal architecture, we first identify its source model. If there are other models in the pool that share exactly the same architectural configuration at this layer, parameter merging (such as Task Arithmetic or TIES Merging) is performed among these isomorphic models. If the architecture of this layer is unique in the pool, the merging step is skipped, and the parameters from the selected source model are used directly. For example, if a given layer is selected from Llama-2-7B, and the pool also contains WizardMath-7B or CodeLlama-7B (which share the same architecture), parameter merging is conducted among these models for the corresponding layer; otherwise, the parameters from Llama-2-7B alone are used. Therefore, the upper-level parameter merging does not rely on unified parameter initialization across all candidate models. Instead, it adaptively decides whether to merge parameters or simply use those from a single model based on the architectural compatibility of each layer. This design allows HM3 to flexibly handle models with heterogeneous architectures within the same merging framework.
> > > We agree with your observation that our approach is fundamentally different from traditional parameter-level model merging methods. Conventional methods typically merge models by adjusting parameters at each layer within a fixed architecture. In contrast, our work focuses on architecture-level model merging, which breaks the assumptions of LMC, since models with different architectures or training trajectories may not reside on a low-loss connected manifold in parameter space.
> > > The theoretical foundation of our approach is mainly rooted in optimization theory. Specifically, we formulate the joint parameter-architecture merging problem as a unified bi-level optimization task, which can be equivalently transformed into a Stackelberg game. We further prove the existence of an equilibrium for this game, which guarantees that the solution decomposition does not lose optimality (see Lemma 1 and related discussion in our paper). Therefore, although our method does not depend on the linear interpolation assumptions of LMC, it is nevertheless rigorously supported by optimization and game-theoretic analysis.
> > >
> > > Furthermore, our experimental results demonstrate that our method significantly outperforms both classical and state-of-the-art parameter-level merging approaches across multiple LLMs and vision models (see Tables 1–4), indicating that architecture-level merging not only differs in mechanism but also provides practical performance advantages.
> > >
> > > Finally, we sincerely appreciate your attention to the theoretical foundations of architecture-level model merging. We acknowledge that the theoretical analysis in this area is still at an early stage, but empirical evidence already demonstrates its feasibility and effectiveness. In the future, we plan to focus on several theoretical directions for architecture-level model merging: (1) investigating nonlinear or piecewise mode connectivity under structural variations to reveal the reachability and transition paths in parameter space; (2) systematically analyzing how mechanisms such as glue layers guarantee intermediate representation consistency, based on representation alignment and information bottleneck theories; and (3) quantifying the effects of structural merging on generalization error and model expressiveness, as well as exploring how structural changes affect adaptability and transferability through task transfer and inductive bias analysis. We hope that the research community will approach this emerging direction with openness and a spirit of innovation, as the current theoretical work, which is not fully mature, lays a solid foundation for future breakthroughs in theory and methodology.

---

> > > ### Author Response · Authors · 2025-08-07
> > >
> > > **2. **Thanks for your helpful comment. We address your question from two aspects:
> > >
> > > First, regarding the training of the MLP alignment, to construct the training data, we randomly sample 10% of the original validation set and extract intermediate feature activations from each candidate pair of stitched layers. For each such pair, we compute statistical descriptors (mean, variance) of the respective activations. The MLP is then optimized to generate a projection matrix that transforms the source layer's output so that its mean and variance match those of the target layer. The loss function combines mean-squared error terms for first- and second-order statistics and includes a CORAL loss for higher-order feature alignment. In practice, each MLP is lightweight, typically with two hidden layers, and requires only modest data and computation. During architecture search, we train the MLP for each candidate connection for 5~15 epochs, with early stopping if the validation loss ceases to improve. On a single GPU, we evaluate the total training time for all glue layers across a merged model (usually with 10~20 MLP alignments) is within 10–15 minutes. For our practical experiments using four NVIDIA 3090 GPUs, we evaluate that the total wall-clock time is reduced to 3~6 minutes, since the training of different MLP alignments can be parallelized. This process efficiently ensures distributional and dimensional compatibility between stitched layers, with minimal overhead. In the revised manuscript, we will describe the detailed data requirements and conduct experiments to record and report the detailed results about the computational overhead.
> > >
> > > Second, the training of the MLP occurs only during the actor-critic-based architecture search phase. Then, when obtaining the final merged model, there is no additional subsequent training phase or large-scale fine-tuning required for the obtained merged model.
> > >
> > > This is the main difference from our method and other architecture-level merging  (i.e.,SOLAR 10.7B merged by Mixtral-7B via Franken Merging). SOLAR 10.7B needs to continue learning after architecture-level merging, while our method can achieve performance gains through training-free property.
> > >
> > >
> > >
> > > **3. **Thank you for your helpful comment. As clarified in our response to W1/Q1 in the rebuttal, the reward in the HM3 framework is strictly outcome-based. Specifically, the reward function is computed only after the entire inference path (trajectory) is generated and the MLP-based alignment is performed. In the revised manuscript, we explicitly state that the reward $R$ is calculated only after the complete trajectory and alignment step: $R = \left( \sum_{k=1}^{K} \lambda_k f_k(\pmb{\theta}, \pmb{h}) \right) - \beta T$.
> > >
> > > This terminal reward is then **uniformly assigned to all time steps in the trajectory** as follows:
> > >
> > > $R_t = R, \quad \forall\, t \in {1, \ldots, T}$
> > >
> > > where $f_k(\cdot)$ denotes the task-specific evaluation metric and $\beta\,T$ penalizes excessively long paths. This uniform assignment is implemented to facilitate efficient storage of transitions and subsequent updates of the policy and value networks in the actor-critic framework.
> > >
> > > Once again, in the revised manuscript, we updated the reward formula to indicate that reward obtained by model evaluation after obtaining the complete inference path and performing MLP alignment, thus ensuring the rigor and correctness of reward calculation.
> > >
> > >
> > >
> > > **4. **Thanks for your question. Certainly, we are committed to fostering transparency and reproducibility in the research community. As mentioned in our paper’s abstract and experimental section, we intend to publicly release our complete source code, including detailed implementations of the hierarchical multi-objective merging framework, reinforcement learning components (actor-critic networks, policy discretization strategy based on Wolpertinger mapping), upon acceptance.
> > >
> > >
> > >
> > > Once again, we sincerely appreciate your careful review and constructive comments.

---

> > > > ### Comment · Reviewer_B6zw · 2025-08-07
> > > >
> > > > Thank you very much. I hope the author can clarify the difference from general model merging in the final version. I'm very grateful for your efforts and will continue to follow your work. I will raise my score to Accept.

---

> > > > > ### Author Response · Authors · 2025-08-07
> > > > >
> > > > > We sincerely thank you for your valuable feedback and positive evaluation. We greatly appreciate your recognition of our efforts and your constructive suggestions. In the version, we will further clarify the distinction between our method and general model merging approaches, as you suggested. Thank you again for your support and encouragement. We are grateful for your decision and will continue to refine our work.

---

### Note · Authors · 2025-08-13

Thanks for the valuable comments of reviewers. To address their concerns, we made the following major revisions.  We clarified the details of MDP, corrected the reward formula, and described a detailed RL process with full hyperparameters; emphasized HM3’s hierarchical architecture–parameter optimization. We expanded experiments to a new domain (GLUE), new tasks in the current domain (XNLI, MathQA, MBPP), larger models (LLaMA-2-13B), and added PCB and Consensus-Merging as SOTA baselines, showing HM3’s advantage and benefit from integrating their outputs. Hyperparameter tuning strategies and related work on Model Evolver/GENOME were added. For cost evaluation, we compared HM3 to Grid Search and EA under equal budgets, showing superior performance. For stability analysis, we averaged over 10 runs, reported standard deviations, and updated convergence plots. Presentation was improved via typo correction, unified notation, and a notation summary table. On MLP alignment, we added moment-matching analysis and ablations confirming its necessity.

We sincerely thank the Area Chair and the reviewers for their thorough evaluations, constructive feedback, and recognition of HM3’s novelty and methodological contributions. Their detailed comments have been invaluable in helping us improve both the technical depth and the presentation quality of the paper. We appreciate the reviewers’ suggestions and believe the revisions have substantially enhanced the paper. We are grateful for the opportunity to address the raised concerns.

---

### Decision · Program_Chairs · 2025-09-17

**Decision:**

Accept (spotlight)

**Comment:**

This paper presents an interesting model merging method that simultaneously supports parameter-level and architecture-level merges, allowing flexible architectural extensions. All the reviewers vote for acceptance.